# Synthesis of Pyrrolidine Monocyclic Analogues of Pochonicine and Its Stereoisomers: Pursuit of Simplified Structures and Potent β-*N*-Acetylhexosaminidase Inhibition

**DOI:** 10.3390/molecules25071498

**Published:** 2020-03-25

**Authors:** Xin Yan, Yuna Shimadate, Atsushi Kato, Yi-Xian Li, Yue-Mei Jia, George W. J. Fleet, Chu-Yi Yu

**Affiliations:** 1Beijing National Laboratory for Molecular Science (BNLMS), CAS Key Laboratory of Molecular Recognition and Function, Institute of Chemistry, Chinese Academy of Sciences, Beijing 100190, China; yanxin2012@iccas.ac.cn (X.Y.); jiayuemei@iccas.ac.cn (Y.-M.J.); 2University of Chinese Academy of Sciences, Beijing 100049, China; 3Department of Hospital Pharmacy, University of Toyama, 2630 Sugitani, Toyama 930-0194, Japan; m1961224@ems.u-toyama.ac.jp; 4Chemistry Research Laboratory, Department of Chemistry, University of Oxford, Mansfield Road, Oxford OX13TA, UK; george.fleet@chem.ox.ac.uk; 5National Engineering Research Center for Carbohydrate Synthesis, Jiangxi Normal University, Nanchang 330022, China

**Keywords:** Pochonicine, pharmacophore, synthesis, β-*N*-acetylhexosaminidase, iminosugars, structure-activity relationship

## Abstract

Ten pairs of pyrrolidine analogues of pochonicine and its stereoisomers have been synthesized from four enantiomeric pairs of polyhydroxylated cyclic nitrones. Among the ten *N*-acetylamino pyrrolidine analogues, only compounds with 2,5-dideoxy-2,5-imino-d-mannitol (DMDP) and pochonicine (**1**) configurations showed potent inhibition of β-*N*-acetylhexosaminidases (β-HexNAcases); while 1-amino analogues lost almost all their inhibitions towards the tested enzymes. The assay results reveal the importance of the *N*-acetylamino group and the possible right configurations of pyrrolidine ring required for this type of inhibitors.

## 1. Introduction

Since its isolation from fungus *Pochonia suchlasporia* var. *suchlasporia* TAMA 87 in 2009 [1], pochonicine (**1**) (Figure 1) has been an attractive synthetic target due to its potent and specific inhibition of β-*N*-acetylhexosaminidases (β-HexNAcases), including β-*N*-acetylglucosaminidases (β-GlcNAcases) and β-*N*-acetylgalactosaminidases (β-GalNAcases) [1,2]. β-HexNAcases are associated with many crucial biological processes [3]. In fungi and insects, the enzymes play important roles in the metabolism of the polysaccharide chitin [4,5]; in mammals, β-HexNAcases participate in the regulation of cell signalling and influence protein expression, degradation and trafficking [6]. In humans, many diseases including lysosomal storage disorders [7], type-II diabetes [6], insulin resistance [8] and Alzheimer’s disease [9,10,11] can be attributed to abnormality of β-HexNAcases. Therefore, study of β-HexNAcase inhibitors may provide alternate strategies for discovery of therapeutic drugs.

As one of the most potent β-HexNAcase inhibitors, pochonicine (**1**) possesses a polyhydroxylated pyrrolizidine skeleton, in contrast to the polyhydroxylated piperidine ring present in the other two naturally occurring iminosugars, nagstatin (**2**) [12] and siastatin B (**3**) (Figure 1) [13]. The *N*-acetylamino group is the common structural feature that distinguishes these natural products from the other iminosugars [14]. A diverse range of potent synthetic β-HexNAcase iminosugar inhibitors have been reported including pyrrolidines (**4** [15,16,17], **5** [18,19], **6** [20,21] and **7** [22]), piperidines (**8** [23,24], **9** [25] and **10** [26]), azepanes (**11** [27,28,29]) and azetidines (**12** [30] and **13** [31]); almost all of them contain an *N*-acetylamino group (Figure 2) [32].

The novel structure and inhibition properties of pochonicine (**1**) led to the rapid report of total syntheses [2], together with its enantiomer [33] and analogues [34]. However, syntheses of such highly substituted bicyclic pyrrolizidines with up to seven contiguous stereogenic centres are long and complex [35,36]. In contrast, construction of monocyclic pyrrolidine iminosugars with only four chiral centres are easier to accomplish. The pyrrolidine core of the corresponding iminosugar frequently appears to be the HexNAcase pharmacophore [37,38,39], as the pyrrolidine sections are responsible for mimicking the transition-state of enzyme reaction [40]. Among these monocyclic iminosugars, a series of pyrrolidines containing acetamide groups which are essential for their β-HexNAcase inhibitions were reported (Figure 2), among which considerable examples were accomplished before the isolation of pochonicine (**1**) [41,42]. Expectedly, quite a number of these five-membered acetamide derivatives were found to be potent β-HexNAcase inhibitors [16,21,25,26]. In continuation of our interests in structure-activity relationship study of iminosugars [39,43,44,45,46], in this work, 20 stereoisomeric pyrrolidine analogues of pochonicine (**1**) were synthesized and systematically assayed as glycosidase inhibitors, in order to look for novel molecules with simplified structure and remained potent inhibitory activities.

## 2. Results and Discussion

### 2.1. Synthesis of 1-N-Acetylamino-2,5-Imino-1,2,5-Trideoxy-l-Mannitol hydrochloride (A-10)

Synthesis of acetamide modified pyrrolidines generally rely on asymmetric synthesis from achiral starting materials or begin with carbohydrate precursors [32]. In the second strategy, carbohydrate-derived nitrones have significant advantages due to their ready chirality, availability and versatile chemistry [47,48]. In this work, polyhydroxylated nitrones **A**–**H** (Figure 3) were readily prepared from the enantiomers of xylose, arabinose, lyxose and ribose by literature methods [16,49,50,51].

The d-xylose-derived nitrone **A** is a convenient starting material for the synthesis of 1-amino-2,5-imino-1,2,5-trideoxy-l-mannitol hydrochloride (**A-11**) and its 1-*N*-acetylamino derivative (**A-10**) [16]. Nucleophilic additions with trimethylsilyl cyanide (TMSCN) [16] and nitromethane [52] were studied as agents for the introduction of aminomethyl substituents. Reaction of nitrone **A** with TMSCN at room temperature provide hydroxylamine **A-2** in 96% yield as the sole product, and its C-2 configuration was determined as *S*-configuration through NOESY experiments since a strong interaction of H-2 and H-4 was observed; in contrast, the aza-Henry reaction gave a pair of inseparable epimers (**A-2′a** and **A-2′b**) in a 1:1 ratio. Since the corresponding reduction products were also difficult to purify, the attempt to introduce aminomethyl groups by aza-Henry reaction was not further investigated. The addition product **A-2** was treated with Raney Ni/H_2_ in the presence of Boc_2_O, and then deprotected to afford diamine **A-3** in good yield. Final hydrogenation of **A-3** gave 1-amino-2,5-imino-1,2,5-trideoxy-l-mannitol dihydrochloride (**A-11**) in quantitative yield (Scheme 1).

Treatment of diamine **A-3** with acetic anhydride provided compound **A-4** in high yield, but the attempt to release the secondary amine selectively by hydrochloric acid was unsatisfactory, giving the target product **A-5** in only 19% yield together with part of diamine **A-3** recovered (Scheme 2). Selective reduction of the *N*-*O* bond of hydroxylamine **A-2**, can be achieved in the presence of SmI_2_ according to the reported method in moderate yield [53]. Since excessive amount of SmI_2_ was needed in above step, other reduction conditions were also tried. Though the typical Zn-Cu(OAc)_2_-AcOH system [16,54] led to complex mixtures, the modified condition with zinc replaced by iron [55] provided the target amine **A-6** in excellent yield with no effect on the nitrile. Subsequent *N*-protection gave the carbamate **A-7**, which was then hydrogenated to convert the cyanide group to the primary amine. However, the reduction to amine **A-8** proceeded in low yield (20–30%). Though the remaining two steps, *N*-acetylation and deprotection both can go smoothly to afford their corresponding products, the unsatisfactory reduction yield of compound **A-7** seriously reduced the total yield of the whole route. Selective acetylation of the primary amine **A-3** was also tried by strictly control of the usage of acetic anhydride and reaction time [15,16], but no target product was obtained. Unexpectedly, acetylation with acetic acid in dichloromethane provided the monoacetylated compound **A-5** in 18% yield (improved to 24% yield when anhydrous MgSO_4_ was added). However, further attempts to improve the reaction were unsuccessful. In contrast, the mild acylation reagent *N*,*N*′,*N*″,*N*′″-tetraacetylglycoluril [56] showed excellent selectivity for primary amine acetylation, giving the monoacetylated **A-5** in 92% yield when refluxing together in dichloromethane. Final hydrogenation of the intermediate then furnished the target product **A-10** quantitatively.

### 2.2. Synthesis of 1-Amino and 1-N-Acetylamino Modified Pyrrolidine Stereoisomers

According to the strategy in Scheme 1 and Scheme 2, other 18 stereoisomeric pyrrolidine analogues of pochonicine (**1**) were synthesized from the corresponding nitrones **B**−**H**. Reaction of TMSCN with nitrones gave exclusively *trans*-addition products in high yields, with nitrone **E** and **F** as exceptions. Cyanation of nitrone **E** and **F** both afforded a pair of diastereomers with *trans*/*cis* ratios as 63:37 and 61:39, respectively. The configurations of the newly constructed chiral centres in compound **D-2**, **F-2b** and **H-2** were unambiguously confirmed by X-ray crystallographic analysis (See Appendix A). The structures of their corresponding enantiomers **C-2**, **E-2b** and **G-2** can also be confirmed since their NMR data were indeed identical. Reduction of the resulting hydroxylamines by Raney Ni/H_2_ in the presence of Boc_2_O and subsequent deprotection afforded diamines **B-3**−**H-3**, which were then acetylated on the primary amine groups to give compounds **B-5**−**H-5**. Final hydrogenation of intermediates **B-3**−**H-3** and **B-5**−**H-5** provided the target products, i.e., **4·HCl**, **C-10**−**H-10** and **B-11**−**H-11** (Table 1).

### 2.3. Glycosidase Inhibition

The synthesized 1-*N*-acetylamino and 1-amino pyrrolidine analogues were assayed against a range of enzymes, as shown in Table 2 and Table 3.

Compound **D-10** which resembles the pyrrolidine ring of pochonicine (**1**) exhibited potent inhibition of β-GlcNAcases from various resources including bovine liver, HL60 and Jack bean (IC_50_ 2.8 μM, 10 μM and 0.12 μM, respectively). While compound **4·HCl**, the 1-deoxy-1-*N*-acetylamino derivative of 2,5-dideoxy-2,5-imino-d-mannitol (DMDP) [57], also behaved as similar potent inhibitor of bovine liver and Jack bean β-GlcNAcase (IC_50_ 4.7 μM and 0.21 μM, respectively). Both the two compounds were found to potent inhibitors of HL60 β-GalNAcase (IC_50_ 9.5 μM and 8.8 μM, respectively). However, comparing to the natural product pochonicine (**1**), the very powerful inhibitor of β-GlcNAcases and β-GalNAcases, both the two analogues showed significant decrease in the inhibition of two enzymes. Unexpectedly, pochonicine (**1**) did not exhibit any inhibition of α-GalNAcases, but compound **D-10** showed moderate inhibition of chicken liver α-GalNAcase (IC_50_ 65.3 μM). For other 1-*N*-acetylamino compounds tested in Table 2, part of the compounds are only moderate or weak inhibition of the tested β-HexNAcases, and the other completely lost their β-HexNAcase inhibitory activities.

As shown in Table 3, 1-amino pyrrolidine analogues failed to provide positive assay results. Most of them showed weak or no inhibition of all the glycosidases tested. As an exception, compound **D-11** with the pyrrolidine ring of pochonicine (**1**) were found to be moderate inhibitor of α-mannosidase (IC_50_ 54 μM) and β-GlcNAcase (IC_50_ 99 μM) from Jack bean, and α-GalNAcase (IC_50_ 44 μM) from chicken liver. The assay results further indicated the importance of acetamide groups for β-HexNAcase inhibitors. However, the more potent inhibition of compound **D-11** of α-GalNAcase than compound **D-10** may indicate the significant role of the scaffold in interaction with active sites of the enzyme instead of the 1-*N*-acetylamino group.

Though all the compounds in Table 2 and Table 3 can also be regarded as 1-*N*-acetylamino and 1-amino derivatives of their corresponding pyrrolidines (for example, compound **4·HCl** is the 1-deoxy-1-*N*-acetylamino derivative of DMDP), they lost almost all their inhibition towards other tested glycosidases including glucosidase, galactosidase, mannosidase, α-l-fucosidase, trehalase, amyloglucosidase, α-l-rhamnosidase and β-glucuronidase, revealing the importance of C-1 hydroxyl groups in interaction with the corresponding enzymes.

Therefore, both configurations of the pyrrolidine ring and the 1-*N*-acetylamino group have significant influences on the inhibition of β-HexNAcases and α-GalNAcase. In detail, 1-*N*-acetylamino pyrrolidine analogues with the same configuration as DMDP and pochonicine (**1**) showed powerful inhibition of these enzymes, revealing the importance of the right configurations of A ring. Furthermore, the results indicate that pochonicine analogue with the A ring in DMDP configuration may also turn out to be potent inhibitors of the above enzymes. The structure-activity relationship reported in this work may be helpful in pursuing simplified pochonicine (**1**) analogues and more potent glycosidase inhibitors.

## 3. Materials and Methods

### 3.1. General Methods

All reagents were used as received without any further purification or prepared as described in the literature. TLC plates were visualized by ultraviolet light or by treatment with a spray of Pancaldi reagent ((NH_4_)_6_MoO_4_, Ce(SO_4_)_2_, H_2_SO_4_, H_2_O) or a 0.5% solution of KMnO_4_ in acetone. Column chromatography was performed on a flash column chromatography with silica gel (200–300 mesh, Inno-chem, Beijing, China). NMR spectra were measured in CDCl_3_ (with TMS as internal standard) or D_2_O (with H_2_O as internal standard) on a Bruker AV300, AV400 or AV500 magnetic resonance spectrometer (Bruker, Ettlingen, Germany) (^1^H NMR at 300 MHz, 400 MHz or 500 MHz, ^13^C NMR at 125 MHz). High-resolution mass spectra (HRMS) were performed on a Thermo Fisher Exactive Spectrometer (Thermo Fisher Scientific, Waltham, MA, USA). Polarimetry was determined using an Optical Activity AA-10R polarimeter with concentrations (*c*) given in gram per 100 mL. Infrared spectra were recorded as films on KBr plates on a Nicolet-6700 FT-IR spectrometer (Thermo Fisher Scientific, Waltham, MA, USA).

### 3.2. Material and Methods for the Enzyme Inhibition Assay

With rat intestinal maltase as an exception, other enzymes were purchased from Sigma-Aldrich Chemical Co. (St. Louis, MO, USA). Brush border membranes prepared from rat small intestine according to the method of Kessler et al. [58] were assayed at pH 6.8 for rat intestinal maltase using maltose. The released d-glucose was determined colorimetrically using the Glucose CII-test Wako (Wako Pure Chemical Ind.; Osaka, Japan). Other glycosidase activities were determined using an appropriate *p*-nitrophenyl glycoside as substrate in a buffer solution at the optimal pH value of each enzyme. The reaction was stopped by adding 400 mM Na_2_CO_3_. The released *p*-nitrophenol was measured spectrometrically at 400 nm [59].

### 3.3. Chemistry

#### 3.3.1. General Procedure for Synthesis of Hydroxylamines **A-2**, **B-2**, **C-2**, **D-2**, **E-2a**, **E-2b**, **F-2a**, **F-2b**, **G-2** and **H-2** with **A-2** as an Example

To a solution of nitrone **A** (1.25 g, 3.00 mmol) in THF (5 mL) and methanol (25 mL) was added dropwise TMSCN (0.45 mL, 3.60 mmol) under Ar atmosphere at 0 °C. After stirring at room temperature for 6–8 h, TLC showed completion of the reaction. The resulting solution was concentrated *in vacuo*, and the residue was purified by flash column chromatography (silica gel, petroleum ether/EtOAc = 6/1) to give hydroxylamine **A-2** (colourless syrup, 1.28 g, 96% yield). Data for **(2*S*,3*S*,4*S*,5*S*)-3,4-bis(benzyloxy)-5-(benzyloxymethyl)-2-cyano-1-hydroxypyrrolidine (A-2)** (Ref. [16]): [α]_D_^23^ −9.5 (*c* 1.1 in CH_2_Cl_2_); *ν*_max_/cm^−1^: 3366 (s), 3088 (m), 3063 (m), 3030 (s), 2920 (s), 2867 (vs), 2237 (w), 1497 (s), 1454 (vs), 1362 (s), 1207 (m), 1100 (vs), 1028 (s), 738 (vs), 697 (vs); ^1^H NMR (500 MHz, CDCl_3_) *δ* (ppm): 7.36–7.20 (m, 15H), 6.52 (s, 1H, OH), 4.54–4.37 (m, 6H), 4.20 (d, *J* = 1.5 Hz, 1H, H-2), 4.13 (t, *J* = 2.1 Hz, 1H, H-3), 3.93 (dd, *J* = 6.5 and 2.2 Hz, 1H, H-4), 3.72 (dd, *J* = 10.5 and 3.3 Hz, 1H, H-6), 3.54 (dd, *J* = 10.5 and 4.0 Hz, 1H, H-6′), 3.28–3.25 (m, 1H, H-5); ^13^C NMR (125 MHz, CDCl_3_) *δ* (ppm): 137.4, 137.3, 136.4, 128.7, 128.4, 128.3, 128.0, 127.97, 127.93, 127.88, 127.82, 115.7, 83.6 (C-3), 81.2 (C-4), 73.3 (Ph*CH_2_*), 72.3 (Ph*CH_2_*), 72.0 (Ph*CH_2_*), 69.4 (C-5), 66.4 (C-6), 61.2 (C-2); HRMS (ESI): calcd for C_27_H_28_O_4_N_2_Na^+^ [M + Na^+^] 467.1941, found 467.1941.

Data for **(2*R*,3*R*,4*R*,5*R*)-3,4-bis(benzyloxy)-5-(benzyloxymethyl)-2-cyano-1-hydroxypyrrolidine (B-2)**: Colourless syrup, 1.26 g, 94% yield from nitrone **B** (1.26 g, 3.02 mmol); [α]_D_^22^ +8.7 (*c* 1.0 in CH_2_Cl_2_); *ν*_max_/cm^−1^: 3359 (m), 3031 (m), 2868 (m), 2239 (w), 1496 (m), 1454 (s), 1362 (m), 1207 (m), 1097 (vs), 1027 (m), 737 (s), 697 (vs); ^1^H NMR (500 MHz, CDCl_3_) *δ* (ppm): 7.36–7.20 (m, 15H), 6.46 (s, 1H, OH), 4.55–4.37 (m, 6H), 4.20 (d, d, *J* = 1.5 Hz, 1H, H-2), 4.14 (t, *J* = 2.1 Hz, 1H, H-3), 3.94 (dd, *J* = 6.5 and 2.2 Hz, 1H, H-4), 3.72 (dd, *J* = 10.5 and 3.3 Hz, 1H, H-6), 3.54 (dd, *J* = 10.5 and 4.0 Hz, 1H, H-6′), 3.28–3.26 (m, 1H, H-5); ^13^C NMR (125 MHz, CDCl_3_) *δ* (ppm): 137.4, 137.3, 136.4, 128.7, 128.4, 128.3, 128.0, 127.96, 127.93, 127.87, 127.82, 115.7, 83.6 (C-3), 81.2 (C-4), 73.3 (Ph*CH_2_*), 72.3 (Ph*CH_2_*), 72.0 (Ph*CH_2_*), 69.4 (C-5), 66.4 (C-6), 61.1 (C-2); HRMS (ESI): calcd for C_27_H_28_O_4_N_2_Na^+^ [M + Na^+^] 467.1941, found 467.1937.

Data for **(2*S*,3*S*,4*R*,5*S*)-3,4-bis(benzyloxy)-5-(benzyloxymethyl)-2-cyano-1-hydroxypyrrolidine (C-2)** (Ref. [60]): Colourless syrup, 1.32 g, 96% yield from nitrone **C** (1.30 g, 3.12 mmol); [α]_D_^23^ +8.1 (*c* 1.5 in CH_2_Cl_2_); *ν*_max_/cm^−1^: 3306 (m), 3030 (m), 2925 (s), 2855 (s), 2251 (w), 1497 (m), 1454 (s), 1362 (m), 1209 (m), 1144 (s), 1102 (s), 1027 (s), 736 (s), 697 (vs); ^1^H NMR (500 MHz, CDCl_3_) *δ* (ppm.): 7.34–7.23 (m, 15H), 6.72 (s, 1H, OH), 4.69–4.44 (m, 6H), 4.33 (t, *J* = 5.5 Hz, 1H, H-3), 4.28 (d, *J* = 5.6 Hz, 1H. H-2), 4.22 (t, *J* = 5.7 Hz, 1H, H-4), 3.72 (dd, *J* = 9.6 Hz and 6.9 Hz, 1H, H-6), 3.66 (dd, *J* = 9.3 Hz and 6.7 Hz, 1H, H-6′), 3.55–3.51 (m, 1H, H-5); ^13^C NMR (125 MHz, CDCl_3_) *δ* (ppm): 137.79, 137.70, 136.7, 128.6, 128.45, 128.43, 128.2, 128.0, 127.9, 127.89, 127.81, 116.4, 80.2 (C-3), 75.8 (C-4), 73.8 (Ph*CH_2_*), 73.4 (Ph*CH_2_*), 73.1 (Ph*CH_2_*), 68.9 (C-5), 67.4 (C-6), 60.4 (C-2); HRMS (ESI): calcd for C_27_H_28_O_4_N_2_Na^+^ [M + Na^+^] 467.1941, found 467.1939.

Data for **(2*R*,3*R*,4*S*,5*R*)-3,4-bis(benzyloxy)-5-(benzyloxymethyl)-2-cyano-1-hydroxypyrrolidine (D-2)** (Ref. [61]): Colourless syrup, 1.19 g, 92% yield from nitrone **D** (1.22 g, 2.93 mmol); [α]_D_^23^ −10.2 (*c* 1.0 in CH_2_Cl_2_); *ν*_max_/cm^−1^: 3328 (m), 3030 (m), 2925 (s), 2870 (s), 2251 (w), 1497 (m), 1454 (s), 1362 (m), 1209 (m), 1145 (s), 1102 (s), 1027 (s), 736 (s), 697 (vs); ^1^H NMR (500 MHz, CDCl_3_) *δ* (ppm): 7.36–7.22 (m, 15H), 6.78 (s, 1H, OH), 4.69–4.43 (m, 6H), 4.33 (t, *J* = 5.5 Hz, 1H, H-3), 4.28 (d, *J* = 5.6 Hz, 1H. H-2), 4.22 (t, *J* = 5.7 Hz, 1H, H-4), 3.72 (dd, *J* = 9.6 Hz and 6.9 Hz, 1H, H-6), 3.65 (dd, *J* = 9.3 Hz and 6.7 Hz, 1H, H-6′), 3.55–3.51 (m, 1H, H-5); ^13^C NMR (125 MHz, CDCl_3_) *δ* (ppm): 137.79, 137.71, 136.7, 128.6, 128.46, 128.43, 128.2, 128.0, 127.89, 127.81, 116.5, 80.2 (C-3), 75.8 (C-4), 73.8 (Ph*CH_2_*), 73.4 (Ph*CH_2_*), 73.1 (Ph*CH_2_*), 69.0 (C-5), 67.4 (C-6), 60.4 (C-2); HRMS (ESI): calcd for C_27_H_28_O_4_N_2_Na^+^ [M + Na^+^] 467.1941, found 467.1938.

**(2*R*,3*R*,4*R*,5*S*)-3,4-bis(benzyloxy)-5-(benzyloxymethyl)-2-cyano-1-hydroxypyrrolidine (E-2a) and (2*S*,3*R*,4*R*,5*S*)-3,4-bis(benzyloxy)-5-(benzyloxymethyl)-2-cyano-1-hydroxypyrrolidine (E-2b)**: 92% total yield from nitrone E (1.34 g, 3.21 mmol).

Data for **E-2a**: White solid, 821 mg, 58% yield; m.p. 117–119 °C; [α]_D_^23^ +7.8 (*c* 0.7 in CH_2_Cl_2_); *ν*_max_/cm^−1^: 3308 (m), 3028 (m), 2870 (m), 2245 (w), 1497 (m), 1453 (m), 1361 (m), 1144 (m), 1109 (s), 1043 (m), 1028 (m), 736 (vs), 693 (vs); ^1^HNMR (500 MHz, CDCl_3_) *δ* (ppm):7.37–7.23 (m,15H), 5.75 (s, 1H, OH), 4.56–4.45 (m, 6H), 4.15 (dd, *J* = 5.6 and 1.4 Hz, 1H, H-2), 4.00 (dd, *J* = 6.1 and 1.4 Hz, 1H, H-4), 3.83 (t, *J* = 9.1 Hz,1H), 3.73 (m, 2H, H-6), 3.39–3.35 (m,1H, H-5); ^13^C NMR (125 MHz, CDCl_3_) *δ* (ppm): 137.8, 137.2, 136.4, 128.6, 128.5, 128.4, 128.3, 128.0, 127.91, 127.90, 127.83, 127.81, 118.35, 83.9 (C-3), 80.0 (C-4), 73.5 (Ph*CH_2_*), 72.4 (Ph*CH_2_*), 72.3 (Ph*CH_2_*), 69.1 (C-6), 67.1 (C-5), 63.2 (C-2); HRMS (ESI): calcd for C_27_H_28_O_4_N_2_Na^+^ [M + Na^+^] 467.1941, found 467.1938.

Data for **E-2b**: White solid, 491 mg, 35% yield; m.p. 111–114 °C; [α]_D_^23^ −38.1 (*c* 1.0 in CH_2_Cl_2_); *ν*_max_/cm^−1^: 3359 (m), 3031 (m), 2868 (m), 2237 (w), 1496 (m), 1454 (m), 1362 (m), 1207 (m), 1097 (s), 1027 (m), 737 (vs), 697 (vs); ^1^H NMR (500 MHz, CDCl_3_) *δ* (ppm):7.35–7.21 (m, 15H), 5.98 (s, 1H, OH), 4.59–4.48 (m, 6H), 4.37 (d, *J* = 6.4 Hz, 1H, H-2), 4.18 (dd, *J* = 7.7 and 4.3 Hz, 1H, H-4), 4.07 (dd, *J* = 6.3 and 4.4 Hz, 1H, H-3), 3.80 (dd, *J* = 9.7 and 5.7 Hz, 1H, H-6), 3.68 (dd, *J* = 9.6 and 5.7 Hz, 1H, H-6′), 3.63−3.59 (m, 1H, H-5); ^13^C NMR (125 MHz, CDCl_3_) *δ* (ppm): 137.7, 137.5, 136.6, 128.6, 128.4, 128.3, 128.2, 127.9, 127.8, 127.78, 127.73, 114.6, 81.1 (C-3), 80.8 (C-4), 73.5 (Ph*CH_2_*), 73.0 (Ph*CH_2_*), 72.9 (Ph*CH_2_*), 67.4 (C-6), 66.7 (C-5), 60.6 (C-2); HRMS (ESI): calcd for C_27_H_28_O_4_N_2_Na^+^ [M + Na^+^] 467.1941, found 467.1939.

**(2*S*,3*S*,4*S*,5*R*)-3,4-bis(benzyloxy)-5-(benzyloxymethyl)-2-cyano-1-hydroxypyrrolidine (F-2a) and (2*R*,3*S*,4*S*,5*R*)-3,4-bis(benzyloxy)-5-(benzyloxymethyl)-2-cyano-1-hydroxypyrrolidine (F-2b)**: 96% total yield from nitrone F (1.50 g, 3.60 mmol). 

Data for **F-2a**: White solid, 985 mg, 59% yield; m.p. 118–119 °C; [α]_D_^23^ −5.5 (*c* 1.0 in CH_2_Cl_2_); *ν*_max_/cm^−1^: 3326 (m), 3028 (m), 2911 (m), 2870 (m), 2245 (w), 1497 (m), 1454 (s), 1241 (m), 1216 (m), 1144 (s), 1110 (vs), 736 (vs), 693 (vs); ^1^H NMR (500 MHz, CDCl_3_) *δ* (ppm):7.37–7.23 (m,15H), 5.70 (s, 1H, OH), 4.56–4.45 (m, 6H), 4.15 (dd, *J* = 5.7 and 1.4 Hz, 1H, H-2), 4.00 (dd, *J* = 6.1 and 1.4 Hz, 1H, H-4), 3.83 (dd, *J* = 9.2 and 7.7Hz,1H, H-3), 3.75–3.71 (m, 2H, H-6), 3.39–3.35 (m,1H); ^13^C NMR (125 MHz, CDCl_3_) *δ* (ppm): 137.8, 137.2, 136.4, 128.6, 128.5, 128.4, 128.3, 128.0, 127.91, 127.90, 127.83, 127.81, 118.3, 83.9 (C-3), 80.0 (C-4), 73.5 (Ph*CH_2_*), 72.4 (Ph*CH_2_*), 72.3 (Ph*CH_2_*), 69.1 (C-6), 67.1 (C-5), 63.2 (C-2); HRMS (ESI): calcd for C_27_H_28_O_4_N_2_Na^+^ [M + Na^+^] 467.1941, found 467.1938.

Data for **F-2b**: White solid, 612 mg, 37% yield; m.p. 113–115 °C; [α]_D_^23^ +35.3 (*c* 1.0 in CH_2_Cl_2_); *ν*_max_/cm^−1^: 3359 (m), 3031 (m), 2868 (m), 2239 (w), 1496 (m), 1454 (m), 1362 (m), 1207 (s), 1097 (s), 1027 (m), 737 (vs), 697 (vs); ^1^H NMR (500 MHz, CDCl_3_) *δ* (ppm):7.35–7.21 (m, 15H), 6.04 (s, 1H, OH), 4.59 –4.47 (m, 6H), 4.37 (d, *J* = 6.3 Hz, 1H, H-2), 4.18 (dd, *J* = 7.7 and 4.3 Hz, 1H, H-4), 4.07 (dd, *J* = 6.3 and 4.4 Hz, 1H, H-3), 3.80 (dd, *J* = 9.7 and 5.7 Hz, 1H, H-6), 3.68 (dd, *J* = 9.6 and 5.7 Hz, 1H, H-6′), 3.63 −3.59 (m, 1H, H-5); ^13^C NMR (125 MHz, CDCl_3_) *δ* (ppm): 137.7, 137.5, 136.6, 128.6, 128.4, 128.3, 128.2, 127.9, 127.8, 127.78, 127.74, 114.6, 81.1 (C-3), 80.8 (C-4), 73.5 (Ph*CH_2_*), 73.0 (Ph*CH_2_*), 72.9 (Ph*CH_2_*), 67.4 (C-6), 66.7 (C-5), 60.6 (C-2); HRMS (ESI): calcd for C_27_H_28_O_4_N_2_Na^+^ [M + Na^+^] 467.1941, found 467.1939.

Data for **(2*R*,3*R*,4*S*,5*S*)-3,4-bis(benzyloxy)-5-(benzyloxymethyl)-2-cyano-1-hydroxypyrrolidine (G-2)** (Ref. [61]): White solid, 1.39 g, 87% yield from nitrone **G** (1.51 g, 3.62 mmol); m.p. 106–109 °C; [α]_D_^23^ +0.6 (*c* 1.0 in CH_2_Cl_2_); *ν*_max_/cm^−1^: 3403 (m), 3029 (m), 2922 (m), 2871 (m), 2242 (w), 1497 (m), 1453 (m), 1352 (m), 1224 (m), 1145 (s), 1101 (s), 1027 (s), 742 (s), 694 (vs); ^1^HNMR (500 MHz, CDCl_3_) *δ* (ppm):7.35–7.24 (m,15H), 6.05 (s, 1H, OH), 4.70–4.43 (m, 6H), 4.10–4.05 (m, 2H, H-3, H-2), 3.83 (t, *J* = 6.1 Hz, 1H, H-4), 3.53–3.51 (m, 2H, H-6, H-6′), 3.31−3.30 (m,1H, H-5); ^13^C NMR (125 MHz, CDCl_3_) *δ* (ppm): 137.6, 137.4, 136.7, 128.6, 128.48, 128.44, 128.2, 128.13, 128.11, 127.98, 127.90, 127.88, 127.86, 118.5, 77.3 (C-3), 75.2 (C-4), 73.3 ((Ph*CH_2_*)), 72.8 (Ph*CH_2_*), 72.2 (Ph*CH_2_*), 72.0 (C-5), 68.0 (C-6), 61.8 (C-2); HRMS (ESI): calcd for C_27_H_28_O_4_N_2_Na^+^ [M + Na^+^] 467.1941, found 467.1936.

Data for **(2*S*,3*S*,4*R*,5*R*)-3,4-bis(benzyloxy)-5-(benzyloxymethyl)-2-cyano-1-hydroxypyrrolidine (H-2)**: White solid, 1.45 g, 91% yield from nitrone **H** (1.50 g, 3.60 mmol); m.p. 109–111 °C; [α]_D_^23^ −1.4 (*c* 1.1 in CH_2_Cl_2_); *ν*_max_/cm^−1^: 3407 (m), 3029 (m), 2873 (m), 2242 (w), 1497 (m), 1453 (m), 1352 (m), 1225 (m), 1149 (s), 1102 (s), 1035 (s), 1028 (s), 743 (vs), 694 (vs); ^1^H NMR (500 MHz, CDCl_3_) *δ* (ppm): 7.34–7.23 (m,15H), 6.11 (s, 1H, OH), 4.68–4.43 (m, 6H), 4.10–4.05 (m, 2H, H-3, H-2), 3.81 (t, *J*= 5.1 Hz, 1H, H-4), 3.53–3.48 (m, 2H, H-6, H-6′), 3.31–3.28 (m,1H, H-5); ^13^C NMR (125 MHz, CDCl_3_) *δ* (ppm): 137.6, 137.4, 136.7, 128.6, 128.49, 128.45, 128.2, 128.13, 128.11, 127.98, 127.91, 127.8, 118.6, 77.3 (C-3), 75.2 (C-4), 73.3 (Ph*CH_2_*), 72.8 (Ph*CH_2_*), 72.2 (Ph*CH_2_*), 72.0 (C-5), 68.0 (C-6), 61.8 (C-2); HRMS (ESI): calcd for C_27_H_28_O_4_N_2_Na^+^ [M + Na^+^] 467.1941, found 467.1933.

#### 3.3.2. General Procedure for Synthesis of Hydroxylamines **A-3**, **B-3**, **C-3**, **D-3**, **E-3a**, **E-3b**, **F-3a**, **F-3b**, **G-3** and **H-3** with **A-3** as an Example

Hydroxylamine **A-2** (1.28 g, 2.88 mmol) was dissolved in methanol (10 mL), followed by Boc_2_O (1.46 mL, 6.30 mmol) and Raney Ni (1.0 g, 60%). The suspension was stirred under hydrogen atmosphere for 4 h when TLC showed completion of the reaction. Hydrogen was replaced by nitrogen, and catalyst was removed from the reaction mixture by filtration. The filtrate was concentrated *in vacuo* to afford the crude product as a colourless oil. The intermediate was dissolved in dichloromethane (10 mL) and cooled to 0 °C, trifluoroacetic acid (0.54 mL, 7.20 mmol) was added dropwise. After stirring at room temperature for 3 h, TLC showed completion of the reaction. The mixture was neutralized by aqueous NaHCO_3_ and extracted with dichloromethane (3 × 10 mL). The organic phases were combined, washed with brine and dried over MgSO_4_. After concentrated *in vacuo*, the crude product was purified by flash column chromatography (silica gel, dichloromethane/methanol = 25:1) to give diamine **A-3** (colourless syrup, 1.05 g, 85% yield for two steps). Data for **(2*S*,3*S*,4*S*,5*S*)-2-(aminomethyl)-3,4-bis(benzyloxy)-5-(benzyloxymethyl)pyrrolidine** (**A-3**): [α]_D_^23^ −16.3 (*c* 1.0 in CH_2_Cl_2_); *ν*_max_/cm^−1^: 3031 (m), 2923 (s), 2865 (s), 1682 (vs), 1453 (m), 1203 (vs), 1130 (vs), 1027 (m), 737 (s), 721 (s), 697 (s); ^1^H NMR (500 MHz, CDCl_3_) *δ* (ppm): 7.34–7.24 (m, 15H), 4.55–4.49 (m, 6H), 3.91 (t, *J* = 3.8 Hz, 1H), 3.76 (t, *J* = 3.8 Hz, 1H), 3.56−3.50 (m, 2H), 3.32 (q, *J* = 5.0 Hz, 1H), 3.15–3.12 (m, 1H), 2.79–2.71 (m, 2H), 1.77 (s, 3H, NH, NH_2_); ^13^C NMR (125 MHz, CDCl_3_) *δ* (ppm): 138.2, 138.16, 138.15, 128.44, 128.41, 127.8, 127.78, 127.75, 127.71, 127.6, 87.3, 86.1, 73.2, 71.9, 71.7, 70.3, 64.1, 61.6, 44.3; HRMS (ESI): calcd for C_27_H_33_O_3_N_2_^+^ [M + H^+^] 433.2486, found 433.2478.

Data for **(2*R*,3*R*,4*R*,5*R*)-2-(aminomethyl)-3,4-bis(benzyloxy)-5-(benzyloxymethyl)pyrrolidine (B-3)**: Colourless syrup, 380 mg, 88% yield from hydroxylamine **B-2** (450 mg, 1.01 mmol); [α]_D_^23^ +11.8 (*c* 0.8 in CH_2_Cl_2_); *ν*_max_/cm^−1^: 3030 (m), 2866 (m), 1682 (vs), 1496 (m), 1453 (m), 1204 (vs), 1130 (vs), 1027 (m), 736 (s), 720 (s), 697 (s); ^1^H NMR (500 MHz, CDCl_3_) *δ* (ppm): 7.34–7.24 (m, 15H), 4.55–4.49 (m, 6H), 3.91 (t, *J* = 3.8 Hz, 1H), 3.76 (t, *J* = 3.8 Hz, 1H), 3.56–3.50 (m, 2H), 3.32 (q, *J* = 5.0 Hz, 1H), 3.16–3.12 (m, 1H), 2.79–2.71 (m, 2H), 1.76 (s, 3H, NH, NH_2_); ^13^C NMR (125 MHz, CDCl_3_) *δ* (ppm): 138.2, 138.16, 138.15, 128.44, 128.41, 127.8, 127.78, 127.75, 127.71, 127.6, 87.3, 86.1, 73.2, 71.9, 71.7, 70.3, 64.1, 61.6, 44.3; HRMS (ESI): calcd for C_27_H_33_O_3_N_2_^+^ [M + H^+^] 433.2486, found 433.2478.

Data for **(2*S*,3*S*,4*R*,5*S*)-2-(aminomethyl)-3,4-bis(benzyloxy)-5-(benzyloxymethyl)pyrrolidine (C-3)**: Colourless syrup, 372 mg, 86% yield from hydroxylamine **C-2** (450 mg, 1.01 mmol); [α]_D_^22^ +13.3 (*c* 1.0 in CH_2_Cl_2_); *ν*_max_/cm^−1^: 3328 (m), 3030 (m), 2864 (m), 1495 (m), 1453 (s), 1143 (s), 1094 (s), 1027 (m), 736 (s), 696 (vs); ^1^H NMR (500 MHz, CDCl_3_) *δ* (ppm): 7.32–7.25 (m, 15H), 4.73–4.43 (m, 6H), 4.04 (t, *J* = 4.1 Hz, 1H), 3.70–3.61 (m, 3H), 3.42 (q, *J* = 6.6 Hz, 1H), 3.30 (q, *J* = 6.3 Hz, 1H), 2.77 (dd, *J* = 12.6 and 4.2 Hz, 1H), 2.59 (dd, *J* = 12.6 and 6.5 Hz, 1H), 1.50 (s, 3H, NH, NH_2_); ^13^C NMR (125 MHz, CDCl_3_) *δ* (ppm): 138.6, 138.2, 138.1, 128.44, 128.41, 128.3, 127.8, 127.78, 127.76, 127.6, 127.5, 81.9, 78.0, 73.3, 73.2, 72.3, 69.7, 61.9, 59.0, 45.2; HRMS (ESI): calcd for C_27_H_33_O_3_N_2_^+^ [M + H^+^] 433.2486, found 433.2479.

Data for **(2*R*,3*R*,4*S*,5*R*)-2-(aminomethyl)-3,4-bis(benzyloxy)-5-(benzyloxymethyl)pyrrolidine (D-3)**: Colourless syrup, 372 mg, 86% yield from hydroxylamine **D-2** (450 mg, 1.01 mmol); [α]_D_^23^ −19.6 (*c* 1.2 in CH_2_Cl_2_); *ν*_max_/cm^−1^: 3292 (w), 3029 (m), 2859 (m), 1585 (m), 1495 (m), 1453 (s), 1143 (s), 1094 (s), 1027 (m), 734 (s), 696 (vs); ^1^H NMR (500 MHz, CDCl_3_) *δ* (ppm): 7.32–7.25 (m, 15H), 4.73–4.43 (m, 6H), 4.04 (t, *J* = 4.1 Hz, 1H), 3.70–3.61 (m, 3H), 3.42 (q, *J* = 6.6 Hz, 1H), 3.30 (q, *J* = 6.3 Hz, 1H), 2.77 (dd, *J* = 12.6 and 4.2 Hz, 1H), 2.59 (dd, *J* = 12.6 and 4.3 Hz, 1H), 1.50 (s, 3H, NH, NH_2_); ^13^C NMR (125 MHz, CDCl_3_) *δ* (ppm): 138.6, 138.2, 138.1, 128.44, 128.41, 128.3, 127.8, 127.78, 127.76, 127.6, 127.5, 81.9, 78.0, 73.3, 73.2, 72.3, 69.7, 61.9, 59.0, 45.2; HRMS (ESI): calcd for C_27_H_33_O_3_N_2_^+^ [M + H^+^] 433.2486, found 433.2478.

Data for **(2*R*,3*R*,4*R*,5*S*)-2-(aminomethyl)-3,4-bis(benzyloxy)-5-(benzyloxymethyl)pyrrolidine (E-3a)**: Colourless syrup, 379 mg, 88% yield from hydroxylamine **E-2a** (450 mg, 1.01 mmol); [α]_D_^23^ +12.1 (*c* 1.0 in CH_2_Cl_2_); *ν*_max_/cm^−1^: 3062 (w), 3029 (m), 2923 (s), 2860 (s), 1688 (m), 1496 (m), 1453 (s), 1201 (m), 1093 (vs), 735 (vs), 697 (vs); ^1^H NMR (500 MHz, CDCl_3_) *δ* (ppm): 7.35–7.24 (m, 15H), 4.57–4.44 (m, 6H), 3.92 (dd, *J* = 4.6 and 1.1 Hz, 1H), 3.73 (dd, *J* = 9.2 and 6.0 Hz, 1H), 3.65 (dd, *J* = 4.5 and 1.1 Hz,1H), 3.62 (dd, *J* = 9.2 and 6.8 Hz, 1H), 3.52 (q, *J* = 5.95 Hz, 1H), 3.12–3.08 (m,1H), 2.85 (dd,12.6 and 4.9 Hz,1H), 2.76 (dd, *J* = 12.6 and 7.1Hz, 1H), 1.61 (s, 3H, NH, NH_2_); ^13^C NMR (125 MHz, CDCl_3_) *δ* (ppm): 138.3, 138.2, 138.0, 128.48, 128.41, 128.3, 127.8, 127.79, 127.75, 127.72, 127.69, 127.63, 127.60, 85.8, 82.9, 73.4, 71.67, 71.64, 69.7, 66.0, 60.6, 45.5; HRMS (ESI): calcd for C_27_H_33_O_3_N_2_^+^ [M + H^+^] 433.2486, found 433.2477.

Data for **(2*S*,3*R*,4*R*,5*S*)-2-(aminomethyl)-3,4-bis(benzyloxy)-5-(benzyloxymethyl)pyrrolidine (E-3b)**: Colourless syrup, 387 mg, 90% yield from hydroxylamine **E-2b** (450 mg, 1.01 mmol); [α]_D_^23^ −27.1 (*c* 0.9 in CH_2_Cl_2_); *ν*_max_/cm^−1^: 3062 (w), 3029 (m), 2920 (s), 2861 (s), 1495 (m), 1453 (s), 1362 (m), 1094 (vs), 1027 (m), 735 (vs), 697 (vs); ^1^H NMR (500 MHz, CDCl_3_) *δ* (ppm): 7.34–7.25 (m, 15H), 4.57–4.38 (m, 6H), 4.04 (dd, *J* = 4.6 and 2.3 Hz, 1H), 3.96 (dd, *J* = 5.3 and 2.3 Hz,1H), 3.67–3.60 (m, 2H), 3.55 (dd, *J* = 8.6 and 6.3 Hz,1H), 3.36 (q, *J* = 6.1 Hz,1H), 2.86–2.78 (m, 2H), 1.63 (s, 3H, NH, NH_2_); ^13^C NMR (125 MHz, CDCl_3_) *δ* (ppm): 138.3, 138.2, 138.1, 128.4, 128.39, 128.35, 127.78, 127.74, 127.71, 127.6, 127.57, 127.56, 82.9, 82.5, 73.3, 72.2, 71.9, 69.5, 61.1, 58.5, 42.1; HRMS (ESI): calcd for C_27_H_33_O_3_N_2_^+^ [M + H^+^] 433.2486, found 433.2479.

Data for **(2*S*,3*S*,4*S*,5*R*)-2-(aminomethyl)-3,4-bis(benzyloxy)-5-(benzyloxymethyl)pyrrolidine (F-3a):** Colourless syrup, 359 mg, 83% yield from hydroxylamine **F-2a** (450 mg, 1.01 mmol); [α]_D_^23^ −9.3 (*c* 1.0 in CH_2_Cl_2_); *ν*_max_/cm^−1^: 3087 (w), 3029 (m), 2922 (s), 2859 (s), 1688 (m), 1496 (m), 1453 (s), 1363 (m), 1201 (s), 1094 (vs), 736 (vs), 697 (vs); ^1^H NMR (500 MHz, CDCl_3_) *δ* (ppm):7.35–7.24 (m, 15H), 4.58–4.44 (m, 6H), 3.92 (dd, *J* = 4.6 and 1.2 Hz, 1H), 3.73 (dd, *J* = 9.2 and 6.0 Hz, 1H), 3.65 (dd, *J* = 4.1 and 1.2 Hz,), 3.62 (dd, *J* = 9.2 and 6.8 Hz, 1H), 3.54–3.50 (m, 1H), 3.12–3.08 (m, 1H), 2.84 (dd, *J* = 12.6 and 4.9 Hz,1H), 2.76 (dd, *J* = 12.6 and 7.1Hz, 1H), 1.60 (s, 3H, NH, NH_2_); ^13^C NMR (125 MHz, CDCl_3_) *δ* (ppm): 138.3, 138.2, 138.0, 128.48, 128.41, 128.3, 127.8, 127.79, 127.72, 127.69, 127.63, 127.60, 85.8, 82.9, 73.4, 71.69, 71.64, 69.7, 66.0, 60.6, 45.5; HRMS (ESI): calcd for C_27_H_33_O_3_N_2_^+^ [M + H^+^] 433.2486, found 433.2478.

Data for **(2*R*,3*S*,4*S*,5*R*)-2-(aminomethyl)-3,4-bis(benzyloxy)-5-(benzyloxymethyl)pyrrolidine (F-3b)**: Colourless syrup, 367 mg, 85% yield from hydroxylamine **F-2b** (450 mg, 1.01 mmol); [α]_D_^22^ +23.8 (*c* 1.0 in CH_2_Cl_2_); *ν*_max_/cm^−1^: 3062 (w), 3029 (m), 2922 (s), 2858 (s), 1495 (m), 1453 (s), 1362 (m), 1093 (vs), 1027 (m), 734 (s), 696 (vs); ^1^H NMR (500 MHz, CDCl_3_) *δ* (ppm): 7.34–7.25 (m, 15H), 4.57–4.38 (m, 6H), 4.04 (dd, *J* = 4.6 and 2.3 Hz, 1H), 3.96 (dd, *J* = 5.3 and 2.3 Hz, 1H), 3.67–3.60 (m, 2H), 3.55 (dd, *J* = 8.6 and 6.3 Hz, 1H), 3.36 (q, *J* = 6.1Hz, 1H), 2.86–2.78 (m, 2H), 1.62 (s, 3H, NH, NH_2_); ^13^C NMR (125 MHz, CDCl_3_) *δ* (ppm): 138.3, 138.2, 138.1, 128.47, 128.42, 128.3, 127.8, 127.77, 127.74, 127.69, 127.60, 127.5, 82.9, 82.5, 73.4, 72.3, 71.9, 69.6, 61.1, 58.6, 42.1; HRMS (ESI): calcd for C_27_H_33_O_3_N_2_^+^ [M + H^+^] 433.2486, found 433.2478.

Data for **(2*R*,3*R*,4*S*,5*S*)-2-(aminomethyl)-3,4-bis(benzyloxy)-5-(benzyloxymethyl)pyrrolidine (G-3)**: Colourless syrup, 375 mg, 87% yield from hydroxylamine **G-2** (450 mg, 1.01 mmol); [α]_D_^22^ +9.2 (*c* 1.1 in CH_2_Cl_2_); *ν*_max_/cm^−1^: 3362 (w), 3061 (m), 3029 (m), 2859 (s), 1496 (m), 1453 (s), 1363 (m), 1094 (vs), 1027 (m), 735 (vs), 697 (vs); ^1^H NMR (400 MHz, CDCl_3_) *δ* (ppm): 7.34–7.22 (m, 15H), 4.60–4.42 (m, 6H), 3.75 (t, *J* = 5.0 Hz, 1H), 3.55 (t, *J* = 5.9 Hz,1H), 3.49 (q, *J* = 5.0 Hz, 1H), 3.46–3.34 (m, 2H), 3.30 (dd, *J* = 10.8 and 5.9 Hz,1H), 2.78 (dd, *J* = 12.8 and 4.4 Hz, 1H), 2.60 (dd, *J* = 12.8 and 6.1 Hz,1H), 1.45 (s, 3H, NH, NH_2_); ^13^C NMR (125 MHz, CDCl_3_): *δ* (ppm): 138.31, 138.30, 138.27, 128.4, 128.3, 128.1, 127.9, 127.7, 127.69, 127.66, 79.4, 78.6, 73.2, 72.1, 71.7, 71.5, 62.9, 61.1, 44.9; HRMS (ESI): calcd for C_27_H_33_O_3_N_2_^+^ [M + H^+^] 433.2486, found 433.2479.

Data for **(2*S*,3*S*,4*R*,5*R*)-2-(aminomethyl)-3,4-bis(benzyloxy)-5-(benzyloxymethyl)pyrrolidine (H-3)**: Colourless syrup, 380 mg, 88% yield from hydroxylamine **H-2** (450 mg, 1.01 mmol); [α]_D_^23^ −15.4 (*c* 1.0 in CH_2_Cl_2_); *ν*_max_/cm^−1^: 3061 (m), 3029 (m), 2861 (s), 1496 (m), 1453 (s), 1362 (m), 1122 (s), 1094 (s), 1027 (m), 736 (s), 697 (vs); ^1^H NMR (400 MHz, CDCl_3_) *δ* (ppm): 7.34–7.23 (m, 15H), 4.60–4.43 (m, 6H), 3.75 (t, *J* = 5.0 Hz, 1H), 3.55 (t, *J* = 5.9 Hz,1H), 3.49 (dd, *J* = 9.9 and 5.0 Hz, 1H), 3.46–3.34 (m, 2H), 3.30 (dd, *J* = 10.8 and 5.9 Hz,1H), 2.78 (dd, *J* = 12.8 and 4.4 Hz, 1H), 2.60 (dd, *J* = 12.8 and 6.1 Hz, 1H), 1.43 (s, 3H, NH, NH_2_); ^13^C NMR (125 MHz, CDCl_3_) *δ* (ppm): 138.3, 138.29, 138.26, 128.4, 128.3, 128.1, 127.9, 127.7, 127.69, 127.65, 79.4, 78.6, 73.2, 72.1, 71.7, 71.5, 62.9, 61.1, 44.9; HRMS (ESI): calcd for C_27_H_33_O_3_N_2_^+^ [M + H^+^] 433.2486, found 433.2479.

#### 3.3.3. Synthesis of **(2*S*,3*S*,4*S*,5*S*)-3,4-bis(benzyloxy)-5-(benzyloxymethyl)-2-cyano-pyrrolidine (A-6)**

To a suspension of iron powder (560 mg, 10.00 mmol, used as received) in acetic acid was added Copper (II) acetate (20 mg, 0.10 mmol), and the mixture was stirred at room temperature for 5–10 min until the bluish green suspension turned into reddish brown. The solution of hydroxylamine **A-2** (450 mg, 1.01 mmol) in acetic acid (10 mL) was added, and the reaction mixture was stirred at room temperature overnight. Solvent was removed *in vacuo*, the residue was neutralized by aqueous NaHCO_3_ and filtered. The resulting filtrate was extracted with EtOAc (3 × 50 mL), then organic phases were combined, dried over MgSO_4_ and concentrated under reduced pressure. Purification by flash chromatography on silica gel (petroleum ether/EtOAc = 3/1) afforded pyrrolidine **A-6** (light yellow syrup, 389 mg, 91% yield). Data for **(2*S*,3*S*,4*S*,5*S*)-3,4-bis(benzyloxy)-5-(benzyloxymethyl) -2-cyano-pyrrolidine (A-6)**: [α]_D_^23^ −21.2 (*c* 1.0 in CH_2_Cl_2_); *ν*_max_/cm^−1^: 3293 (w), 3030 (w), 2926 (m), 2869 (m), 2246 (w), 1717 (m), 1661 (s), 1453 (s), 1397(m), 1149 (s), 1102 (s), 1027 (m), 736 (s), 697 (vs); ^1^H NMR (500 MHz, CDCl_3_) *δ* (ppm): 7.36–7.25 (m, 15H), 4.59–4.45 (m, 6H), 4.27 (t, *J* = 3.1 Hz, 1H), 3.76 (d, *J* = 2.9 Hz, 1H), 3.83 (dd, *J* = 5.5 and 3.3 Hz, 1H), 3.59 (q, *J* = 6.5 Hz, 1H), 3.49–3.45 (m, 2H), 2.45 (br, 1H, NH); ^13^C NMR (125 MHz, CDCl_3_) *δ* (ppm): 137.7, 1377.5, 136.7, 128.6, 128.4, 128.2, 128.0, 127.87, 127.82, 127.78, 127.74, 119.3, 87.4, 84.1, 73.3, 72.5, 72.2, 69.7, 62.1, 51.9; HRMS (ESI): calcd for C_27_H_29_O_3_N_2_^+^ [M + H^+^] 429.2173, found 429.2181.

#### 3.3.4. Synthesis of ***tert*-butyl-(2*S*,3*S*,4*S*,5*S*)-2-(aminomethyl)-3,4-bis(benzyloxy)-5-(benzyloxy methyl)pyrrolidine-1-carboxylate (A-8)**

The mixture of compound **A-6** (389 mg, 0.91 mmol) and Et_3_N (190 μL, 1.37 mmol) in dichloromethane (5 mL) was cooled by an ice-water bath, and Boc_2_O (298 mg, 1.37 mmol) was added. After stirring overnight at room temperature, the reaction was quenched by aqueous NaHCO_3_. The solution was then extracted with dichloromethane (3 × 10 mL), then organic phases were combined, dried over MgSO_4_ and concentrated *in vacuo* to give intermediate **A-7** as a light yellow syrup (479 mg, 99% yield). The crude **A-7** was directly dissolved in methanol (5 mL), and Raney Ni (500 mg, 60%) was added. The suspension was stirred under hydrogen atmosphere for 24 h when TLC showed part of intermediate **A-7** remained unreacted. Longer reaction time did not lead to any further change. Hydrogen was then replaced by nitrogen, and catalyst was removed from the reaction mixture. The filtrate was concentrated *in vacuo* to afford a colourless oil, which was purified by flash chromatography (silica gel, dichloromethane/methanol = 50:1) to give compound **A-8** (colourless syrup, 126 mg, 26% yield). Data for ***tert*-butyl-(2*S*,3*S*,4*S*,5*S*)-2-(aminomethyl)-3,4-bis(benzyloxy)-5 -(benzyloxymethyl)pyrrolidine-1-carboxylate (A-8)**: [α]_D_^23^ −12.7 (*c* 0.6 in CH_2_Cl_2_); *ν*_max_/cm^−1^: 3029 (w), 2973 (m), 2929 (m), 1688 (s), 1453 (m), 1391 (vs), 1367 (s), 1160 (s), 1108 (s), 1027 (m), 734 (s), 697 (s); ^1^H NMR (500 MHz, CDCl_3_) *δ* (ppm): 7.37–7.23 (m, 15H), 4.77–4.41 (m, 6H), 4.18 (d, *J* = 7.7 Hz, 1H), 4.15–4.10 (m, 2H), 3.99–3.94 (m, 1H), 3.94–3.87 (m, 1H), 3.57 (d, *J* = 6.5 Hz,1H), 3.24 (d, *J* = 13.0 Hz, 1H), 2.87 (dd, *J* = 12.9 and 8.5 Hz, 1H), 1.39 (s, 9H); ^13^C NMR (125 MHz, CDCl_3_) *δ* (ppm): 156.7, 138.3, 137.8, 137.7, 128.38, 128.33, 128.2, 127.89, 127.84, 127.69, 127.60, 127.5, 127.4, 82.1, 80.3, 77.2, 73.2, 72.4, 72.1, 69.5, 60.3, 59.2, 44.4, 28.2; HRMS (ESI): calcd for C_32_H_40_O_5_N_2_^+^ [M + H^+^] 533.3010, found 533.3013.

#### 3.3.5. Synthesis of ***tert*-butyl-(2*S*,3*S*,4*S*,5*S*)-2-*N*-acetylaminomethyl-3,4-bis(benzyloxy)-5-(benzyloxymethyl)pyrrolidine-1-carboxylate (A-9)**

Compound **A-8** (126 mg, 0.24 mmol) was dissolved in dichloromethane (5 mL), followed by Ac_2_O (28 μL, 0.29 mmol) and catalytic amount of DMAP. The solution was stirred at room temperature for 3-4 h, when TLC showed completion of the reaction. The solution was quenched by aqueous NaHCO_3_, and extracted with dichloromethane (3 × 10 mL). The organic phases were combined, dried over MgSO_4_ and concentrated *in vacuo*. The residue was purified by flash chromatography (silica gel, petroleum ether/EtOAc = 1/1) to afford compound **A-9** (colourless syrup, 124 mg, 93% yield). Data for ***tert*-butyl-(2*S*,3*S*,4*S*,5*S*)-2-*N*-acetylaminomethyl-3,4-bis(benzyloxy)-5-(benzyloxy methyl)pyrrolidine-1-carboxylate (A-9)**: [α]_D_^23^ −19.1 (*c* 1.0 in CH_2_Cl_2_); *ν*_max_/cm^−1^: 3299 (w), 3029 (w), 2927 (m), 1690 (vs), 1682 (m), 1453 (m), 1390 (s), 1366 (s), 1274 (m), 1174 (m), 1096 (s), 1027 (m), 735 (m), 697 (m); ^1^H NMR (500 MHz, CDCl_3_) *δ* (ppm): 7.33–7.25 (m, 15H), 6.64 (br, 1H, NHCO), 4.75–4.44 (m, 6H), 4.18 (dd, *J* = 8.8 and 6.1 Hz, 1H), 4.15–4.09 (m, 2H), 3.95–3.85 (m, 2H), 3.57 (d, *J* = 6.5 Hz,1H), 3.46–3.37 (m, 1H), 3.19–3.10 (m, 1H), 1.87 (s, 1H), 1.39 (s, 9H); ^13^C NMR (125 MHz, CDCl_3_) *δ* (ppm): 170.3, 155.5, 138.4, 138.1, 128.3, 128.2, 127.8, 127.7, 127.68, 127.62, 127.4, 80.7, 79.8, 77.8, 73.2, 72.5, 72.4, 70.0, 61.7, 58.6, 43.0, 28.3, 23.2; HRMS (ESI): calcd for C_34_H_43_O_6_N_2_^+^ [M + H^+^] 575.3116, found 575.3118.

#### 3.3.6. General Procedure for Synthesis of Compounds **A-5**, **B-5**, **C-5**, **D-5**, **E-5a**, **E-5b**, **F-5a**, **F-5b**, **G-5** and **H-5** with **A-5** as an Example

To the solution of compound **A-3** (215 mg, 0.50 mmol) in dichloromethane (10 mL) was added *N*,*N*’,*N*’’,*N*’’’-tetraacetylglycoluril (170 mg, 0.55 mmol). The solution was refluxed for 3–5 h, when TLC showed disapearance of the starting material. The reaction was quenched by aqueous NaHCO_3_, and extracted with dichloromethane (3 × 10 mL). The organic phases were combined, dried over MgSO_4_ and concentrated under reduced pressure. The crude product was purified by flash chromatography (silica gel, dichloromethane/methanol = 50:1) to give compound **A-5** (colourless syrup, 208 mg, 88% yield). Data for **(2*S*,3*S*,4*S*,5*S*)-2-*N*-acetylaminomethyl-3,4-bis(benzyloxy)-5- (benzyloxymethyl)pyrrolidine (A-5)**: [α]_D_^23^ −7.7 (*c* 0.9 in CH_2_Cl_2_); *ν*_max_/cm^−1^: 3292 (w), 3030 (m), 2860 (m), 1656 (s), 1453 (m), 1363 (m), 1094 (s), 1027 (m), 737 (s), 697 (vs); ^1^H NMR (500 MHz, CDCl_3_) *δ* (ppm): 7.34–7.25 (m, 15H), 6.08 (br, 1H, NHCO), 4.55–4.49 (m, 6H), 3.88 (t, *J* = 3.5 Hz, 1H), 3.76 (t, *J* = 3.4 Hz, 1H), 3.54–3.48 (m, 2H), 3.43–3.38 (m, 1H), 3.37–3.31 (m, 2H), 3.28–3.23 (m, 1H), 2.1 (br, 1H, NH), 1.88 (s, 3H); ^13^C NMR (125 MHz, CDCl_3_) *δ* (ppm): 170.2, 137.9, 137.83, 137.81, 128.46, 128.44, 127.87, 127.84, 127.81, 127.77, 127.73, 86.8, 85.5, 73.2, 72.0, 71.8, 69.8, 61.9, 61.0, 41.4, 23.2; HRMS (ESI): calcd for C_29_H_35_O_4_N_2_^+^ [M + H^+^] 475.2591, found 475.2588.

Data for **(2*R*,3*R*,4*R*,5*R*)-2-*N*-acetylaminomethyl-3,4-bis(benzyloxy)-5-(benzyloxymethyl) pyrrolidine (B-5)** (Ref. [62]): Colourless syrup, 216 mg, 91% yield from diamine **B-3** (215 mg, 0.50 mmol); [α]_D_^23^ +11.9 (*c* 1.0 in CH_2_Cl_2_); *ν*_max_/cm^−1^: 3283 (w), 3030 (m), 2859 (m), 1656 (s), 1454 (m), 1363 (m), 1093 (s), 1028 (m), 737 (s), 697 (vs); ^1^H NMR (500 MHz, CDCl_3_) *δ* (ppm): 7.34–7.25 (m, 15H), 6.08 (br, 1H, NHCO), 4.55–4.49 (m, 6H), 3.88 (t, *J* = 3.5 Hz, 1H), 3.76 (t, *J* = 3.4 Hz, 1H), 3.54–3.48 (m, 2H), 3.43–3.38 (m, 1H), 3.37–3.31 (m, 2H), 3.28–3.23 (m, 1H), 2.16 (br, 1H, NH), 1.88 (s, 3H); ^13^C NMR (125 MHz, CDCl_3_) *δ* (ppm): 170.2, 137.9, 137.83, 137.81, 128.46, 128.44, 127.87, 127.84, 127.81, 127.77, 127.73, 86.8, 85.5, 73.2, 72.0, 71.8, 69.8, 61.9, 61.0, 41.4, 23.2; HRMS (ESI): calcd for C_29_H_35_O_4_N_2_^+^ [M + H^+^] 475.2591, found 475.2586.

Data for **(2*S*,3*S*,4*R*,5*S*)-2-*N*-acetylaminomethyl-3,4-bis(benzyloxy)-5-(benzyloxymethyl) pyrrolidine (C-5)**: Colourless syrup, 201 mg, 85% yield from diamine **C-3** (215 mg, 0.50 mmol); [α]_D_^23^ +11.9 (*c* 1.0 in CH_2_Cl_2_); *ν*_max_/cm^−1^: 3288 (w), 3029 (w), 2924 (m), 2855 (m), 1652 (s), 1453 (m), 1366 (m), 1089 (s), 1027 (m), 736 (s), 697 (s); ^1^H NMR (500 MHz, CDCl_3_) *δ* (ppm): 7.35–7.25 (m, 15H), 5.87 (br, 1H, NH), 4.73–4.49 (m, 6H), 4.00 (t, *J* = 4.0 Hz, 1H), 3.68−3.62 (m, 2H), 3.58 (t, *J* = 7.5 Hz, 1H), 3.44–3.38 (m, 2H), 3.35–3.25 (m, 2H), 2.16 (br, 2H, NH_2_), 1.89 (s, 3H); ^13^C NMR (125 MHz, CDCl_3_) *δ* (ppm): 170.3, 138.3, 138.0, 137.8, 128.47, 128.42, 128.3, 127.89, 127.88, 127.86, 127.80, 127.7, 127.6, 82.4, 77.6, 73.4, 73.2, 72.5, 69.5, 59.07, 59.04, 42.3, 23.2; HRMS (ESI): calcd for C_29_H_35_O_4_N_2_^+^ [M + H^+^] 475.2591, found 475.2588.

Data for **(2*R*,3*R*,4*S*,5*R*)-2-*N*-acetylaminomethyl-3,4-bis(benzyloxy)-5-(benzyloxymethyl) pyrrolidine (D-5)**: Colourless syrup, 194 mg, 82% yield from diamine **D-3** (215 mg, 0.50 mmol); [α]_D_^23^ −2.5 (*c* 1.0 in CH_2_Cl_2_); *ν*_max_/cm^−1^: 3296 (w), 3028 (w), 2921 (m), 2855 (m), 1652 (s), 1554 (m), 1453 (s), 1365(m), 1091 (s), 1027 (m), 736 (s), 696 (s); ^1^H NMR (500 MHz, CDCl_3_) *δ* (ppm): 7.35–7.25 (m, 15H), 5.87 (br, 1H, NHCO), 4.73–4.49 (m, 6H), 4.00 (t, *J* = 4.1 Hz, 1H), 3.68–3.62 (m, 2H), 3.58 (t, *J* = 7.5 Hz, 1H), 3.44–3.38 (m, 2H), 3.35–3.25 (m, 2H), 2.16 (br, 1H, NH), 1.89 (s, 3H); ^13^C NMR (125 MHz, CDCl_3_) *δ* (ppm): 170.3, 138.4, 138.0, 137.9, 128.49, 128.44, 128.3, 127.9, 127.8, 127.7, 127.6, 82.4, 77.7, 73.4, 73.3, 72.6, 69.5, 59.09, 59.06, 42.3, 23.2; HRMS (ESI): calcd for C_29_H_35_O_4_N_2_^+^ [M + H^+^] 475.2591, found 475.2588.

Data for **(2*R*,3*R*,4*R*,5*S*)-2-*N*-acetylaminomethyl-3,4-bis(benzyloxy)-5-(benzyloxymethyl) pyrrolidine (E-5a)**: Colourless syrup, 213 mg, 90% yield from diamine **E-3a** (215 mg, 0.50 mmol); [α]_D_^23^ +9.2 (*c* 1.0 in CH_2_Cl_2_); *ν*_max_/cm^−1^: 3295 (w), 3030 (w), 2924 (m), 2855 (m), 1652 (s), 1554 (m), 1493 (s), 1365 (m), 1091 (s), 1027 (m), 736 (s), 696 (s); ^1^H NMR (500 MHz, CDCl_3_) *δ* (ppm): 7.34–7.24 (m, 15H), 6.81 (br, 1H, NHCO), 4.58–4.42 (m, 6H), 4.19 (br, 1H, NH), 3.90 (d, *J* = 3.9 Hz, 1H), 3.77 (s, 1H), 3.73 (dd, *J* = 9.3 and 5.8 Hz, 1H), 3.68 (dd, *J* = 9.5 and 6.7 Hz, 2H), 3.62 (dd, *J* = 10.3 and 6.0 Hz, 1H), 3.48–3.43 (m, 1H), 3.39–3.34 (m, 1H), 1.76 (s, 3H); ^13^C NMR (125 MHz, CDCl_3_) *δ* (ppm): 170.6, 138.1, 137.7, 137.6, 128.54, 128.52, 128.4, 127.99, 127.91, 127.8, 127.79, 127.75, 127.6, 85.0, 82.2, 73.5, 71.9, 71.8, 69.6, 62.0, 60.4, 42.4, 22.8; HRMS (ESI): calcd for C_29_H_35_O_4_N_2_^+^ [M + H^+^] 475.2591, found 475.2585.

Data for **(2*S*,3*R*,4*R*,5*S*)-2-*N*-acetylaminomethyl-3,4-bis(benzyloxy)-5-(benzyloxymethyl) pyrrolidine (E-5b)**: Colourless syrup, 194 mg, 82% yield from diamine **E-3b** (215 mg, 0.50 mmol); [α]_D_^23^ −17.2 (*c* 1.2 in CH_2_Cl_2_); *ν*_max_/cm^−1^: 3290 (m), 3063 (m), 3030 (m), 2925 (s), 2861 (s), 1651 (vs), 1549 (m), 1453 (s), 1367 (s), 1092 (vs), 1027 (m), 736 (vs), 697 (vs); ^1^H NMR (500 MHz, CDCl_3_) *δ* (ppm): 7.36–7.25 (m, 15H), 6.11 (br, 1H, NHCO), 4.56–4.35 (m, 6H), 3.99 (d, *J* = 1.6 Hz, 1H), 3.93 (d, *J* = 3.1 Hz, 1H), 3.63–3.61 (m, 2H), 3.57–3.52 (m, 3H), 3.30–3.25 (m, 1H), 2.36 (br, 1H, NH), 1.85 (s, 3H); ^13^C NMR (125 MHz, CDCl_3_) *δ* (ppm): 170.4, 138.1, 137.9, 137.7, 128.6, 128.4, 128.0, 127.89, 127.86, 127.84, 127.7, 82.6, 82.0, 73.4, 72.3, 72.1, 68.8, 58.9, 58.2, 39.6, 23.2; HRMS (ESI): calcd for C_29_H_35_O_4_N_2_^+^ [M + H^+^] 475.2591, found 475.2584.

Data for **(2*S*,3*S*,4*S*,5*R*)-2-*N*-acetylaminomethyl-3,4-bis(benzyloxy)-5-(benzyloxymethyl) pyrrolidine (F-5a)**: Colourless syrup, 206 mg, 87% yield from diamine **F-3a** (215 mg, 0.50 mmol); [α]_D_^22^ −4.1 (*c* 1.0 in CH_2_Cl_2_); *ν*_max_/cm^−1^: 3295 (w), 3064 (m), 3030 (m), 2924 (m), 2855 (m), 1652 (s), 1554 (m), 1493 (m), 1365 (m), 1091 (s), 1027 (m), 736 (s), 696 (s); ^1^H NMR (500 MHz, CDCl_3_) *δ* (ppm): 7.34–7.23 (m, 15H) 6.54 (br, 1H, NHCO), 4.58–4.42 (m, 6H), 3.89 (dd, *J* = 4.1 and 0.6 Hz, 1H), 3.74–3.71 (m, 2H), 3.65 (dd, *J* = 8.9 and 6.9 Hz, 1H), 3.59 (dd, *J* = 10.0 and 5.4 Hz, 1H), 3.44–3.40 (m, 2H), 3.35–3.31 (m, 1H), 3.19 (br, 1H, NH), 1.73 (s, 3H); ^13^C NMR (125 MHz, CDCl_3_) *δ* (ppm): 170.6, 138.1, 137.7, 137.6, 128.54, 128.52, 128.4, 127.99, 127.9, 127.8, 127.79, 127.75, 127.6, 85.0, 82.2, 73.5, 71.9, 71.8, 69.6, 62.0, 60.4, 42.4, 22.8; HRMS (ESI): calcd for C_29_H_35_O_4_N_2_^+^ [M + H^+^] 475.2591, found 475.2587.

Data for **(2*R*,3*S*,4*S*,5*R*)-2-*N*-acetylaminomethyl-3,4-bis(benzyloxy)-5-(benzyloxymethyl) pyrrolidine (F-5b)**: Colourless syrup, 197 mg, 83% yield from diamine **F-3b** (215 mg, 0.50 mmol); [α]_D_^22^ +20.5 (*c* 1.0 in CH_2_Cl_2_); *ν*_max_/cm^−1^: 3293 (m), 3063 (m), 3030 (m), 2924 (s), 2861 (s), 1651 (vs), 1549 (m), 1453 (s), 1367 (s), 1092 (vs), 1027 (m), 736 (vs), 697 (vs); ^1^H NMR (500 MHz, CDCl_3_) *δ* (ppm): 7.35–7.24 (m, 15H) 6.24 (br, 1H, NHCO), 4.55–4.35 (m, 6H), 3.98 (dd, *J* = 6.0 and 1.9 Hz, 1H), 3.92 (dd, *J* = 4.9 and 1.9 Hz, 1H), 3.63 (m, 2H), 3.57–3.52 (m, 3H), 3.29–3.23 (m, 1H), 3.12 (br, 1H, NH), 1.84 (s, 3H); ^13^C NMR (125 MHz, CDCl_3_) *δ* (ppm): 170.4, 138.1, 137.9, 137.7, 128.6, 128.48, 128.42, 128.0, 127.89, 127.86, 127.85, 127.7, 82.5, 82.0, 73.4, 72.3, 72.1, 68.9, 58.9, 58.2, 39.6, 23.2; HRMS (ESI): calcd for C_29_H_35_O_4_N_2_^+^ [M + H^+^] 475.2591, found 475.2585.

Data for **(2*R*,3*R*,4*S*,5*S*)-2-*N*-acetylaminomethyl-3,4-bis(benzyloxy)-5-(benzyloxymethyl) pyrrolidine (G-5)**: Colourless syrup, 204 mg, 86% yield from diamine **G-3** (215 mg, 0.50 mmol); [α]_D_^23^ +7.5 (*c* 1.0 in CH_2_Cl_2_); *ν*_max_/cm^−1^: 3293 (m), 3062 (m), 3029 (m), 2924 (s), 2861 (s), 1653 (vs), 1539 (m), 1453 (s), 1365 (m), 1100 (vs), 1027 (m), 736 (s), 697 (vs); ^1^H NMR (500 MHz, CDCl_3_) *δ* (ppm): 7.34–7.24 (m, 15H), 6.12 (br, 1H, NHCO), 4.58–4.45 (m, 6H), 3.77 (t, *J* = 4.7 Hz, 1H), 3.60 (t, *J* = 5.5 Hz, 1H), 3.50 (q, *J* = 4.4 Hz, 1H), 3.47–3.38 (m, 4H), 3.18–3.13 (m, 1H), 2.14 (br, 1H, NH), 1.74 (s, 3H); ^13^C NMR (125 MHz, CDCl_3_) *δ* (ppm): 170.2, 138.0, 137.9, 128.47, 128.40, 128.1, 128.0, 127.86, 127.80, 127.7, 80.0, 78.2, 73.3, 72.0, 71.6, 61.0, 59.6, 42.3, 23.0; HRMS (ESI): calcd for C_29_H_35_O_4_N_2_^+^ [M + H^+^] 475.2591, found 475.2587.

Data for **(2*S*,3*S*,4*R*,5*R*)-2-*N*-acetylaminomethyl-3,4-bis(benzyloxy)-5-(benzyloxymethyl) pyrrolidine (H-5)**: Colourless syrup, 192 mg, 81% yield from diamine **H-3** (215 mg, 0.50 mmol); [α]_D_^23^ −1.6 (*c* 1.2 in CH_2_Cl_2_); *ν*_max_/cm^−1^: 3304 (m), 3062 (m), 3030 (m), 2865 (s), 1651 (vs), 1549 (m), 1453 (s), 1365 (m), 1099 (vs), 1027 (m), 736 (vs), 697 (vs); ^1^H NMR (500 MHz, CDCl_3_) *δ* (ppm): 7.33–7.24 (m, 15H), 6.11 (br, 1H, NHCO), 4.57–4.43 (m, 6H), 3.76 (t, *J* = 4.8 Hz, 1H), 3.59 (t, *J* = 5.6 Hz, 1H), 3.49 (dd, *J* = 8.9 and 4.4 Hz, 1H), 3.47–3.37 (m, 4H), 3.18–3.13 (m, 1H), 2.14 (br, 1H, NH), 1.74 (s, 3H); ^13^C NMR (125 MHz, CDCl_3_): *δ* (ppm): 170.2, 138.0, 137.9, 128.47, 128.40, 128.1, 128.0, 127.86, 127.80, 127.7, 80.0, 78.2, 73.3, 72.0, 71.6, 61.0, 59.6, 42.3, 23.0; HRMS (ESI): calcd for C_29_H_35_O_4_N_2_^+^ [M + H^+^] 475.2591, found 475.2588.

#### 3.3.7. General Procedure for Synthesis of 1-*N*-Acetylamino Derivatives (**A-10**, **4·HCl**, **C-10**, **D-10**, **E-10a**, **E-10b**, **F-10a**, **F-10b**, **G-10** and **H-10**) and 1-Amino Derivatives (**A-11**, **B-11**, **C-11**, **D-11**, **E-11a**, **E-11b**, **F-11a**, **F-11b**, **G-11** and **H-11**) with **A-10** as an Example

To a stirred solution of **A-5** (95 mg, 0.20 mmol) and 3 N HCl (0.5 mL) in MeOH (10 mL) was added Pd/C (10 wt%, 30 mg) under Ar atmosphere and the reaction mixture was stirred under H_2_ atmosphere for 8 h. Then the catalyst was filtered and the solvent was removed under reduced pressure to afford compound **A-10** (colourless syrup, 47 mg, 99% yield). Data for **1-*N*-acetylamino-2,5-imino-1,2,5-trideoxy-l-mannitol hydrochloride (A-10)** (Ref. [16], reported as free base): [α]_D_^23^ −30.4 (*c* 0.5 in MeOH); *ν*_max_/cm^−1^: 3290 (vs), 2932 (s), 1646 (m), 1550 (m), 1369 (m), 1132 (m), 1041 (m); ^1^H NMR (500 MHz, D_2_O) *δ* (ppm): 4.25 (s, 1H,), 4.11 (s, 1H), 3.98 (dd, *J* = 12.0 Hz and 4.6 Hz, 1H), 3.95–3.91 (m, 1H), 3.86 (dd, *J* = 11.8 Hz and 8.7 Hz, 1H), 3.66 (d, *J* = 6.8 Hz, 2H), 3.63–3.61 (m, 1H), 2.04 (s, 3H, COCH_3_); ^13^C NMR (125 MHz, D_2_O): *δ* (ppm): 175.2, 75.7, 74.6, 67.5, 61.5, 59.2, 35.8, 21.7; HRMS (ESI): calcd for C_8_H_17_O_4_N_2_^+^ [M + H^+^] 205.1183, found 205.1181.

Data for **1-*N*-acetylamino-2,5-imino-1,2,5-trideoxy-d-mannitol hydrochloride (4·HCl)** (Ref. [16,62], reported as free base): Colourless syrup, 50 mg, 99% yield from compound **B-5** (101 mg, 0.21 mmol); [α]_D_^25^ +34.6 (*c* 0.5 in MeOH); *ν*_max_/cm^−1^: 3297 (vs), 2937 (s), 1644 (m), 1550 (m), 1369 (m), 1132 (m), 1042 (m); ^1^H NMR (500 MHz, D_2_O) *δ* (ppm): 4.22 (d, *J* = 2.0 Hz, 1H), 4.09 (t, *J* = 1.6 Hz, 1H), 3.95 (dd, *J* = 12.1 Hz and 4.8 Hz, 1H), 3.91 (dt, *J* = 6.9 Hz and 3.7 Hz, 1H), 3.83 (dd, *J* = 12.1 Hz and 8.6 Hz, 1H), 3.66 (d, *J* = 6.8 Hz, 2H), 3.61–3.58 (m, 1H), 2.01 (s, 3H, COCH_3_); ^13^C NMR (125 MHz, D_2_O) *δ* (ppm): 175.2, 75.7, 74.6, 67.5, 61.5, 59.2, 35.8, 21.7; HRMS (ESI): calcd for C_8_H_17_O_4_N_2_^+^ [M + H^+^] 205.1183, found 205.1183.

Data for **1-*N*-acetylamino-2,5-imino-1,2,5-trideoxy-l-altritol hydrochloride (C-10)** (Ref. [17], reported as free base): Colourless syrup, 45 mg, 99% yield from compound **C-5** (90 mg, 0.19 mmol); [α]_D_^22^ −18.0 (*c* 0.7 in MeOH); *ν*_max_/cm^−1^: 3291 (vs), 2936 (s), 1635 (m), 1550 (m), 1419 (m),1369 (m), 1136 (m), 1042 (m); ^1^H NMR (500 MHz, D_2_O) *δ* (ppm): 4.33(s, 1H), 4.23 (dd, *J* = 9.3 Hz and 3.7 Hz, 1H), 3.99 (dd, *J* = 12.0 Hz and 4.8 Hz, 1H), 3.90 (dd, *J* = 11.6 Hz and 8.4 Hz, 1H), 3.82–3.80 (m, 1H), 3.72–3.66 (m, 2H), 3.59 (dd, *J* = 15.3 Hz and 7.7 Hz, 1H), 2.04 (s, 3H, COCH_3_); ^13^C NMR (125 MHz, D_2_O) *δ* (ppm): 176.0, 72.5, 69.8, 62.0, 60.5, 57.4, 38.9, 21.6; HRMS (ESI): calcd for C_8_H_17_O_4_N_2_^+^ [M + H^+^] 205.1183, found 205.1186.

Data for **1-*N*-acetylamino-2,5-imino-1,2,5-trideoxy-d-altritol hydrochloride (D-10)** (Ref. [16,61], reported as free base): Colourless syrup, 41 mg, 99% yield from compound **D-5** (82 mg, 0.17 mmol); [α]_D_^22^ +21.3 (*c* 0.4 in MeOH); *ν*_max_/cm^−1^: 3296 (vs), 2936 (s), 1635 (m), 1550 (m), 1419 (m),1371 (m), 1136 (m), 1042 (m); ^1^H NMR (500 MHz, D_2_O) *δ* (ppm) 4.33 (t, *J* = 3.2 Hz, 1H), 4.23 (dd, *J* = 9.3 Hz and 3.7 Hz, 1H), 3.99 (dd, *J* = 12.0 Hz and 4.8 Hz, 1H), 3.90 (dd, *J* = 11.6Hz and 8.4Hz, 1H), 3.82–3.80 (m, 1H), 3.72–3.66 (m, 2H), 3.59 (dd, *J* = 15.3 Hz and 7.75 Hz, 1H), 2.04 (s, 3H, COCH_3_); ^13^C NMR (125 MHz, D_2_O) *δ* (ppm): 176.0, 72.5, 69.8, 62.0, 60.5, 57.4, 38.9, 21.6; HRMS (ESI): calcd for C_8_H_17_O_4_N_2_^+^ [M + H^+^] 205.1183, found 205.1185.

Data for **1-*N*-acetylamino-2,5-imino-1,2,5-trideoxy-d-glucitol hydrochloride (E-10a)** (Ref. [63], reported as free base): Colourless syrup, 46 mg, 99% yield from compound **E-5a** (91 mg, 0.19 mmol); [α]_D_^25^ +36.9 (*c* 0.85 in MeOH); *ν*_max_/cm^−1^: 3292 (vs), 2935 (s), 1645 (m), 1550 (m), 1369 (m), 1132 (m), 1045 (m); ^1^H NMR (400 MHz, D_2_O) *δ* (ppm): 4.27 (s, 1H), 4.12 (s, 1H), 3.98 (d, *J* = 9.4 Hz, 1H), 3.94–3.88 (m, 2H), 3.68–3.57 (m, 3H), 2.01 (s, 3H, COCH_3_); ^13^C NMR (125 MHz, D_2_O) *δ* (ppm):175.8, 76.8, 74.5, 65.1, 63.0, 57.0, 39.4, 21.6; HRMS (ESI): calcd for C_8_H_17_O_4_N_2_^+^ [M + H^+^] 205.1183, found 205.1182.

Data for **1-*N*-acetylamino-2,5-imino-1,2,5-trideoxy-l-iditol hydrochloride (E-10b)** (Ref. [64], reported as free base): Colourless syrup, 52 mg, 99% yield from compound **E-5b** (104 mg, 0.22 mmol); [α]_D_^25^ −9.6 (*c* 1.2 in MeOH); *ν*_max_/cm^−1^: 3292 (vs), 2936 (s), 1651 (m), 1553 (m), 1370 (m), 1136 (m), 1042 (m); ^1^H NMR (500 MHz, D_2_O) *δ* (ppm): 4.35 (d, *J* = 3.3 Hz, 1H), 4.28 (d, *J* = 2.0 Hz, 1H), 4.01–3.95 (m, 3H), 3.87 (dd, *J* = 11.9 and 8.3 Hz, 1H), 3.60 (d, *J* = 6.8 Hz, 2H), 2.01 (s, 3H, COCH_3_); ^13^C NMR (125 MHz, D_2_O) *δ* (ppm): 175.8, 75.5, 74.1, 62.6, 61.3, 57.9, 38.6, 21.6; HRMS (ESI): calcd for C_8_H_17_O_4_N_2_^+^ [M + H^+^] 205.1183, found 205.1180.

Data for **1-*N*-acetylamino-2,5-imino-1,2,5-trideoxy-l-glucitol hydrochloride (F-10a)**: Colourless syrup, 44 mg, 99% yield from compound **F-5a** (88 mg, 0.19 mmol); [α]_D_^25^ −29.2 (*c* 1.05 in MeOH); *ν*_max_/cm^−1^: 3285 (vs), 2931 (s), 1645 (m), 1551 (m), 1370 (m), 1090 (m), 1063 (m); ^1^H NMR (400 MHz, D_2_O) *δ* (ppm): 4.30 (t, *J* = 1.6 Hz, 1H), 4.16 (t, *J* = 1.6 Hz, 1H), 4.01 (dd, *J* = 8.8 and 2.8 Hz, 1H), 3.95–3.88 (m, 2H), 3.69–3.59 (m, 3H), 2.04 (s, 3H, COCH_3_); ^13^C NMR (125 MHz, D_2_O) *δ* (ppm):175.8, 76.8, 74.6, 65.2, 63.1, 57.0, 39.4, 21.6; HRMS (ESI): calcd for C_8_H_17_O_4_N_2_^+^ [M + H^+^] 205.1183, found 205.1183.

Data for **1-*N*-acetylamino-2,5-imino-1,2,5-trideoxy-d-iditol hydrochloride (F-10b)**: Colourless syrup, 42 mg, 99% yield from compound **F-5b** (83 mg, 0.17 mmol); [α]_D_^25^ +7.7 (*c* 1.0 in MeOH); *ν*_max_/cm^−1^: 3296 (vs), 2936 (s), 1651 (m), 1553 (m), 1371 (m), 1136 (m), 1041 (m); ^1^H NMR (500 MHz, D_2_O) *δ* (ppm): 4.34 (d, *J* = 3.4 Hz, 1H), 4.27 (t, *J* = 1.8 Hz, 1H), 4.00–3.94 (m, 3H), 3.86 (dd, *J* = 11.6 and 8.3 Hz, 1H), 3.60 (d, *J* = 6.9 Hz, 2H), 2.01 (s, 3H, COCH_3_); ^13^C NMR (125 MHz, D_2_O) *δ* (ppm): 175.8, 75.5, 74.1, 62.6, 61.3, 57.9, 38.6, 21.7; HRMS (ESI): calcd for C_8_H_17_O_4_N_2_^+^ [M + H^+^] 205.1183, found 205.1183.

Data for **1-*N*-acetylamino-2,5-imino-1,2,5-trideoxy-d-allitol hydrochloride (G-10):** Colourless syrup, 46 mg, 99% yield from compound **G-5** (92 mg, 0.19 mmol); [α]_D_^23^ −23.1 (*c* 1.0 in MeOH); *ν*_max_/cm^−1^: 3306 (vs), 2928 (s), 1635 (s), 1551 (s), 1419 (m),1370 (m), 1089 (m), 1063 (m); ^1^H NMR (500 MHz, D_2_O) *δ* (ppm): 4.22–4.19 (m, 2H), 3.93 (dd, *J* = 12.6 Hz and 3.8 Hz, 1H), 3.83 (dd, *J* = 12.6 Hz and 3.8 Hz, 1H), 3.78–3.74 (m, 1H), 3.73–3.66 (m, 2H), 3.57 (dd, *J* = 15.1 Hz and 7.4 Hz, 1H), 2.04 (s, 1H, COCH_3_); ^13^C NMR (125 MHz, D_2_O): *δ*(ppm): 176.1, 71.1, 70.2, 64.2, 62.8, 57.9, 38.5, 21.6; HRMS (ESI): calcd for C_8_H_17_O_4_N_2_^+^ [M + H^+^] 205.1183, found 205.1182.

Data for **1-*N*-acetylamino-2,5-imino-1,2,5-trideoxy-l-allitol hydrochloride (H-10)**: Colourless syrup, 41 mg, 99% yield from compound **H-5** (81 mg, 0.17 mmol); [α]_D_^23^ +27.3 (*c* 0.95 in MeOH); *ν*_max_/cm^−1^: 3296 (vs), 2936 (s), 1651 (s), 1546 (s), 1419 (m), 1372 (m), 1091 (m), 1061 (m); ^1^H NMR (500 MHz, D_2_O) *δ* (ppm): 4.22–4.18 (m, 2H), 3.93 (dd, *J* = 12.5 Hz and 3.6 Hz, 1H), 3.82 (dd, *J* = 12.6 Hz and 3.6 Hz, 1H), 3.78−3.74 (m, 1H), 3.71–3.66 (m, 2H), 3.57 (dd, *J* = 15.1 Hz and 7.4 Hz, 1H), 2.04 (s, 1H, COCH_3_); ^13^C NMR (125 MHz, D_2_O) *δ*(ppm): 176.1, 71.1, 70.2, 64.2, 62.8, 57.9, 38.5, 21.5; HRMS (ESI): calcd for C_8_H_17_O_4_N_2_^+^ [M + H^+^] 205.1183, found 205.1184.

Data for **1-amino-2,5-imino-1,2,5-trideoxy-l-mannitol dihydrochloride (A-11)** (Ref. [61], reported as free base): Colourless syrup, 46 mg, 99% yield from compound **A-3** (86 mg, 0.2 mmol); [α]_D_^23^ −66.7 (*c* 1.0 in MeOH); *ν*_max_/cm^−1^: 3313 (s), 2939 (s), 1115 (m), 1058 (m), 1033 (m), 1014 (m); ^1^H NMR (400 MHz, D_2_O) *δ* (ppm): 4.20–4.12 (m, 2H), 3.98 (dd, *J* = 12.5 Hz and 3.5 Hz, 1H), 3.91–3.81 (m, 2H), 3.74–3.69 (m, 1H), 3.57 (d, *J* = 10.5 Hz, 2H); ^13^C NMR (125 MHz, D_2_O) *δ* (ppm): 76.5, 73.9, 63.4, 58.4, 58.1, 38.7; HRMS (ESI): calcd for C_6_H_15_O_3_N_2_^+^ [M + H^+^] 163.1077, found 163.1077.

Data for **1-amino-2,5-imino-1,2,5-trideoxy-d-mannitol dihydrochloride (B-11)** (Ref. [17]): Colourless syrup, 35 mg, 99% yield from compound **B-3** (66 mg, 0.15 mmol); [α]_D_^23^ −66.7 (*c* 1.0 in MeOH); *ν*_max_/cm^−1^: 3314 (s), 2942 (s), 1113 (m), 1063 (m), 1033 (m), 1014 (m); ^1^H NMR (400 MHz, D_2_O) *δ* (ppm): 4.19–4.12 (m, 2H), 3.98 (dd, *J* = 12.6 Hz and 3.6 Hz, 1H), 3.88 (dd, *J* = 12.5 Hz and 3.6 Hz, 1H), 3.83 (q, *J* = 7.2 Hz, 1H), 3.73–3.69 (m, 1H), 3.56 (d, *J* = 10.5 Hz, 2H); ^13^C NMR (125 MHz, D_2_O) *δ* (ppm): 76.5, 73.9, 63.4, 58.4, 58.1, 38.7; HRMS (ESI): calcd for C_6_H_15_O_3_N_2_^+^ [M + H^+^] 163.1077, found 163.1077.

Data for **1-amino-2,5-imino-1,2,5-trideoxy-l-altritol dihydrochloride (C-11)** (Ref. [17]): Colourless syrup, 39 mg, 99% yield from compound **C-3** (73 mg, 0.17 mmol); [α]_D_^23^ −42.0 (*c* 1.0 in MeOH); *ν*_max_/cm^−1^: 3314 (s), 2923 (s), 1132 (m), 1033 (m), 1014 (m); ^1^H NMR (500 MHz, D_2_O) *δ* (ppm): 4.38 (t, *J* = 3.2 Hz, 1H), 4.34 (dd, *J* = 9.5 Hz and 3.7 Hz, 1H), 4.04–3.97 (m, 1H), 3.97–3.92 (m, 2H), 3.85 (ddd, *J* = 13.8 Hz and 7.8 Hz and 5.9 Hz, 1H), 3.62–3.53 (m, 2H); ^13^C NMR (125 MHz, D_2_O) *δ* (ppm): 74.0, 69.3, 62.9, 57.3, 56.9, 39.0; HRMS (ESI): calcd for C_6_H_15_O_3_N_2_^+^ [M + H^+^] 163.1077, found 163.1078.

Data for **1-amino-2,5-imino-1,2,5-trideoxy-d-altritol dihydrochloride (D-11)** (Ref. [61], reported as free base): Colourless syrup, 37 mg, 99% yield from compound **D-3** (69 mg, 0.2 mmol); [α]_D_^23^ +39.0 (*c* 0.8 in MeOH); *ν*_max_/cm^−1^: 3318 (s), 2923 (s), 1131 (m), 1033 (m), 983 (m); ^1^H NMR (500 MHz, D_2_O) *δ*(ppm): 4.38 (t, *J* = 2.5 Hz, 1H), 4.34 (dd, *J* = 9.5 Hz and 3.7 Hz, 1H), 4.06–4.01 (m, 1H), 3.98–3.93 (m, 2H), 3.86 (dd, *J* = 14.9 Hz and 7.9 Hz, 1H), 3.62–3.53 (m, 2H); ^13^C NMR (125 MHz, D_2_O) *δ* (ppm): 74.0, 69.3, 62.9, 57.3, 56.9, 39.0; HRMS (ESI): calcd for C_6_H_15_O_3_N_2_^+^ [M + H^+^] 163.1077, found 163.1076.

Data for **1-amino-2,5-imino-1,2,5-trideoxy-d-glucitol dihydrochloride (E-11a)** (Ref. [17]): Colourless syrup, 38 mg, 99% yield from compound **E-3a** (71 mg, 0.16 mmol); [α]_D_^23^ +21.6 (*c* 0.5 in MeOH); *ν*_max_/cm^−1^: 3313 (s), 2935 (s), 1118 (m), 1063 (m), 1033 (m); ^1^H NMR (300 MHz, D_2_O) *δ* (ppm): 4.32–4.29 (m, 1H), 4.26 (t, *J* = 2.8 Hz, 1H), 4.02–3.90 (m, 3H), 3.80 (dt, *J* = 6.9 and 2.9 Hz, 1H), 3.54 (d, *J* = 6.8 Hz, 2H); ^13^C NMR (125 MHz, D_2_O) *δ* (ppm): 77.5, 74.1, 64.1, 62.4, 56.9, 39.2; HRMS (ESI): calcd for C_6_H_15_O_3_N_2_^+^ [M + H^+^] 163.1077, found 163.1077.

Data for **1-amino-2,5-imino-1,2,5-trideoxy-l-iditol dihydrochloride (E-11b)** (Ref. [17]): Colourless syrup, 44 mg, 99% yield from compound **E-3b** (81 mg, 0.19 mmol); [α]_D_^23^ +5.5 (*c* 1.0 in MeOH); *ν*_max_/cm^−1^: 3322 (s), 2924 (s), 1131 (m), 1039 (m), 984 (m); ^1^H NMR (500 MHz, D_2_O) *δ* (ppm): 4.46 (d, *J* = 5.1 Hz, 1H), 4.44 (d, *J* = 5.1Hz, 1H), 4.22 (dt, *J* = 6.8 Hz and 3.5 Hz, 1H), 4.12-4.09 (m, 1H), 4.03 (dd, *J* = 12.2 Hz and 4.6Hz, 1H), 3.94 (dd, *J* = 12.2 Hz and 8.8 Hz, 1H), 3.62 (dd, *J* = 13.7 Hz and 6.8 Hz, 1H), 3.55–3.51(dd, *J* = 13.7 Hz and 6.8 Hz, 1H); ^13^C NMR (125 MHz, D_2_O) *δ*(ppm): 74.6, 74.2, 63.8, 58.3, 57.3, 35.9; HRMS (ESI): calcd for C_6_H_15_O_3_N_2_^+^ [M + H^+^] 163.1077, found 163.1076.

Data for **1-amino-2,5-imino-1,2,5-trideoxy-l-glucitol dihydrochloride (F-11a)** (Ref. [61], reported as free base): Colourless syrup, 31 mg, 99% yield from compound **F-3a** (57 mg, 0.13 mmol); [α]_D_^23^ −27.5 (*c* 0.65 in MeOH); *ν*_max_/cm^−1^: 3313 (s), 2935 (s), 1118 (m), 1063 (m), 1033 (m); ^1^H NMR (500 MHz, D_2_O) *δ* (ppm): 4.35 (s, 1H), 4.31 (s, 1H), 4.05–3.96 (m, 3H), 3.88–3.84 (m, 1H), 3.59 (d, *J* = 6.6 Hz, 2H); ^13^C NMR (125 MHz, D_2_O) *δ*(ppm): 77.4, 74.0, 64.1, 62.5, 56.8, 39.2; HRMS (ESI): calcd for C_6_H_15_O_3_N_2_^+^ [M + H^+^] 163.1077, found 163.1076.

Data for **1-amino-2,5-imino-1,2,5-trideoxy-d-iditol dihydrochloride (F-11b)** (Ref. [61], reported as free base): Colourless syrup, 34 mg, 99% yield from compound **F-3b** (62 mg, 0.14 mmol); [α]_D_^23^ −7.3 (*c* 0.3 in MeOH); *ν*_max_/cm^−1^: 3310 (s), 2936 (s), 1131 (m), 1076 (m), 1031 (m); ^1^H NMR (500 MHz, D_2_O) *δ* (ppm): 4.41–4.39 (m, 2H), 4.22 (dt, *J* = 6.7 Hz and 3.5 Hz, 1H), 4.05–3.99 (m, 2H), 3.91 (dd, *J* = 11.8 Hz and 8.5 Hz, 1H), 3.57 (dd, *J* = 13.7 Hz and 6.8 Hz, 1H), 3.47(dd, *J* = 13.7 Hz and 6.8 Hz, 1H); ^13^C NMR (125 MHz, D_2_O) *δ* (ppm): 74.7, 74.4, 63.6, 58.1, 57.4, 36.0; HRMS (ESI): calcd for C_6_H_15_O_3_N_2_^+^ [M + H^+^] 163.1077, found 163.1076.

Data for **1-amino-2,5-imino-1,2,5-trideoxy-d-allitol dihydrochloride (G-11)** (Ref. [61], reported as free base): Colourless syrup, 35 mg, 99% yield from compound **G-3** (65 mg, 0.15 mmol); [α]_D_^23^ +8.1 (*c* 1.0 in MeOH); *ν*_max_/cm^−1^: 3322 (s), 2947 (s), 1122 (m), 1078 (m), 1034 (m), 979 (m); ^1^H NMR (500 MHz, D_2_O) *δ* (ppm): 4.33 (dd, *J* = 7.6 Hz and 5.0 Hz, 1H), 4.29 (dd, *J* = 4.7 Hz and 4.0 Hz, 1H), 3.96 (dd, *J* = 12.2 Hz and 3.5 Hz 1H), 3.92–3.84 (m, 3H) 3.61–3.53 (m, 2H); ^13^C NMR (125 MHz, D_2_O) *δ* (ppm): 72.5, 70.0, 66.0, 58.6, 58.1, 38.4; HRMS (ESI): calcd for C_6_H_15_O_3_N_2_^+^ [M + H^+^] 163.1077, found 163.1077.

Data for **1-amino-2,5-imino-1,2,5-trideoxy-l-allitol dihydrochloride (H-11)** (Ref. [61], reported as free base): Colourless syrup, 36 mg, 99% yield from compound **H-3** (67 mg, 0.16 mmol); [α]_D_^22^ −7.2 (*c* 1.0 in MeOH); *ν*_max_/cm^−1^: 3313 (s), 2951 (s), 1124 (m), 1079 (m), 1034 (m), 980 (m); ^1^H NMR (400 MHz, D_2_O) *δ* (ppm):4.34–4.27 (m, 2H), 3.96 (dd, *J* = 12.0 Hz and 3.4 Hz, 1H), 3.92–3.84 (m, 3H), 3.62–3.52 (m, 2H); ^13^C NMR (125 MHz, D_2_O) *δ* (ppm): 72.5, 70.1, 66.1, 58.7, 58.1, 38.4; HRMS (ESI): calcd for C_6_H_15_O_3_N_2_^+^ [M + H^+^] 163.1077, found 163.1077.

## 4. Conclusions

In summary, a general and efficient synthetic strategy has been developed for the synthesis of 1-*N*-acetylamino and 1-amino pyrrolidine analogues of pochonicine (**1**) with l-*arabino*-nitrone (**A**), d-*arabino*-nitrone (**B**), l-*lyxo*-nitrone (**C**), d-*lyxo*-nitrone (**D**), l-*xylo*-nitrone (**E**), d-*xylo*-nitrone (**F**), l-*ribo*-nitrone (**G**) and d-*ribo*-nitrone (**H**) as the starting materials 4 and 5 steps, respectively. Glycosidase inhibition assays on a range of enzymes showed that 1-*N*-acetylamino pyrrolidine analogues with the same configuration as DMDP and pochonicine (**1**) showed powerful inhibition of β-HexNAcases and moderate inhibition of α-GalNAcase, while the other compounds showed weak or no inhibition of the tested glycosidases. This work has further examined the glycosidase inhibition of pyrrolidine analogues of pochonicine and its stereoisomers, and would be helpful for the study of potent and selective β-HexNAcase inhibitors.

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
