# Peer review of "Synthesis of Pyrrolidine Monocyclic Analogues of Pochonicine and Its Stereoisomers: Pursuit of Simplified Structures and Potent β-N-Acetylhexosaminidase Inhibition"

_molecules, 2020, doi:10.3390/molecules25071498_

Round 1
Reviewer 1 Report
Dear Authors,
I carefully read the manuscript and my opinion is
the manuscript presents the synthesis and biological studies of the derivatives of pochonicine. The amount of the synthesis and biological research presented in the manuscript is impressive.
One minor note: the Authors should explain why the stretching vibration of the cyanic group is not observed in IR spectra.
Sincerely
Author Response
please see the attached file for our reply to reviewer 1's questions.

Reviewer 2 Report
Peer-Review of Molecules Manuscript 739597
The manuscript entitled “Synthesis of Pyrrolidine Monocyclic Analogues of Pochonicine and Its Stereoisomers: Pursuit of Simplified Structures and Potent β-N-Acetylhexosaminidase Inhibition” describes the preparation and biological evaluation of a large library of pochonicine analogs.
The work leading to these compounds is mostly well presented, accurately supported by the included references and supporting information, and conclusions are in line with results obtained.
I only have a few minor comments that I think can improve the submitted manuscript before publication in Molecules is granted.
- The way in which the synthetic procedures are presented can be improved by including all reactions within a single scheme, and separating them as needed.
- In all presented cases, cyanation of the corresponding nitrones added a new asymmetric carbon to the chiral starting materials. The authors state that, in most cases, the reaction produced exclusively a single isomer of the two possible diatereoisomers, but do not describe how these assignments were made, beside the cases in which x-ray crystallography was used.
Author Response
please see the attached file for our reply to reviewer 2's questions.
